# miR-142 regulates the tumorigenicity of human breast cancer stem cells through the canonical WNT signaling pathway

Taichi Isobe[1], Shigeo Hisamori[1†], Daniel J Hogan[2†], Maider Zabala[1], David G Hendrickson[3], Piero Dalerba[1,4¶], Shang Cai[1], Ferenc Scheeren[1], Angera H Kuo[1], Shaheen S Sikandar[1], Jessica S Lam[1], Dalong Qian[1], Frederick M Dirbas[5], George Somlo[6], Kaiqin Lao[7§], Patrick O Brown[2], Michael F Clarke[1*], Yohei Shimono[1*‡]

[1]Institute for Stem Cell Biology and Regenerative Medicine, Stanford University, Stanford, United States; [2]Department of Biochemistry, Howard Hughes Medical Institute, Stanford University School of Medicine, Stanford, United States; [3]Department of Chemical and Systems Biology, Stanford University School of Medicine, Stanford, United States; [4]Department of Medicine, Division of Oncology, Stanford University, Stanford, United States; [5]Department of Surgery, Stanford University School of Medicine, Stanford, United States; [6]City of Hope Cancer Center, Duarte, United States; [7]Applied Biosystems, Foster City, United States

*For correspondence: mfclarke@ stanford.edu (MFC); yshimono@ med.kobe-u.ac.jp (YS)

†These authors equally contributed as second authors

Present address: ‡Division of Molecular and Cellular Biology, Kobe University Graduate School of Medicine, Kobe, Japan; ¶Department of Pathology and Cell Biology, Columbia University, New York, United States; §Genetic Sciences Division, Thermo Fisher Scientific, South San Francisco, United States

**Abstract** MicroRNAs (miRNAs) are important regulators of stem and progenitor cell functions. We previously reported that miR-142 and miR-150 are upregulated in human breast cancer stem cells (BCSCs) as compared to the non-tumorigenic breast cancer cells. In this study, we report that miR-142 efficiently recruits the *APC* mRNA to an RNA-induced silencing complex, activates the canonical WNT signaling pathway in an APC-suppression dependent manner, and activates the expression of miR-150. Enforced expression of miR-142 or miR-150 in normal mouse mammary stem cells resulted in the regeneration of hyperproliferative mammary glands in vivo. Knockdown of endogenous miR-142 effectively suppressed organoid formation by BCSCs and slowed tumor growth initiated by human BCSCs in vivo. These results suggest that in some tumors, miR-142 regulates the properties of BCSCs at least in part by activating the WNT signaling pathway and miR-150 expression.

## Introduction

MicroRNAs (miRNAs) are evolutionarily conserved small non-coding RNAs that regulate the translation of mRNAs. They are recruited to an RNA-induced silencing complex (RISC) and bind to the seed sequence within the 3′ untranslated region (UTR) of target mRNAs, leading to destabilization and/or translational suppression of the target mRNAs (*Bartel, 2009*). The immunopurification (IP) of Argonaute (Ago), a central component of the RISC in the human and mouse, followed by microarray analyses (Ago IP/microarray method) makes it possible to isolate any Ago-associated miRNAs and mRNAs without relying on the mechanism of regulation (i.e. mRNA decay or translational suppression), or sequence conservation, enabling a comprehensive identification of the miRNA-target genes in an unbiased manner. This provides quantitative information about the mRNAs that are regulated by miR-NAs (*Hendrickson et al., 2008*, *2009*).

miRNAs are able to regulate the expression of hundreds of target mRNAs simultaneously and control a variety of cell functions including cell proliferation, stem cell maintenance, and differentiation

**eLife digest** Messenger RNA molecules take the information encoded in a gene's DNA sequence and turn it into instructions for building a protein. However, if certain smaller molecules of RNA—called microRNAs—bind to a messenger RNA molecule, they 'silence' it, which prevents the information in the messenger RNA from being translated to make a protein.

Despite their small size, microRNAs are very powerful. These molecules are able to simultaneously inhibit the translation of hundreds of messenger RNAs and perform many roles, including controlling cell growth and maintaining populations of stem cells. Furthermore, microRNAs have been linked to different aspects of the growth of cancerous cells. Certain microRNAs appear to suppress tumors by regulating the growth of the stem cells found there, while others appear to be 'hyperactive' in cancers—including breast cancer, colon cancer, and blood cancer.

In 2009, researchers compared the amount of microRNA in breast cancer stem cells that are highly capable of forming tumors with the amount in other cancer cells within the same tumor. Amongst other differences, two microRNAs (called miR-142 and miR-150) were found to be hyperactive in human breast cancer stem cells. One of them, miR-142, is known to target a gene called *APC* that inhibits the renewal of normal stem cells. Mutations in the *APC* gene have been linked to colon cancer, and scientists have suggested that the mutations inactivate APC in cancer cells to promote unregulated cell growth. Breast tumors rarely have mutations in the *APC* gene, but Isobe et al. wondered whether microRNAs that target this gene might also promote the growth of these tumor cells.

Isobe et al.—including several of the researchers involved in the 2009 work—show that miR-142 does target the *APC* gene in human breast cancer stem cells, and silences it. With the gene silenced, a cancer-promoting pathway turns on and more miR-150 is made. Increasing the amount of either miR-142 or miR-150 causes excessive cell growth in breast tissue and can form abnormal breast tissue in mice. Reducing the amount of miR-142 in human breast cancer stem cells slows the growth of breast tumors.

Although they only make up a small population of human breast cancer cells, focusing on breast cancer stem cells could uncover the cancer-promoting pathways that are activated in human breast cancers.

(*Lewis et al., 2005*). We previously identified a human breast cancer stem cell (BCSC) population (a CD44$^+$ CD24$^{-/low}$ lineage$^-$ population of human breast cancer cells) that in many human breast tumors is enriched for the ability to drive tumor formation in a mouse xenograft model as compared to the remaining non-tumorigenic cancer cells (NTG cells) within the same breast tumor (*Al-Hajj et al., 2003*). Comprehensive analyses of the expression profile of 466 miRNAs revealed that 37 miRNAs are differentially expressed between the human BCSCs and NTG cells (*Shimono et al., 2009*). Among them, both miR-200c and miR-183 are downregulated in the human BCSCs and suppress the protein expression of the stem cell self-renewal gene, BMI1, and miR-200c suppresses the protein expression of the EMT regulator ZEB1 (*Shimono et al., 2009*; *Wellner et al., 2009*). Enforced expression of miR-200c can strongly suppress the tumor formation driven by human BCSCs and the mammary ducts formation by normal mammary stem cells in vivo, suggesting that miR-200c is a regulator of normal mammary and BCSCs. On the other hand, the expression of miRNAs, such as miR-142, miR-150, and miR-155, are upregulated in human BCSCs (*Shimono et al., 2009*). Among them, miR-155 was originally identified as a product of the oncogenic BIC gene locus in B cell lymphoma (*Eis et al., 2005*). Abnormal proliferation and myelodysplasia are seen when miR-155 expression is sustained in the blood system (*O'Connell et al., 2008*). Furthermore, miR-155 functions as an oncogenic miRNA in various cancers, including leukemia and breast cancers (*Czyzyk-Krzeska and Zhang, 2013*). Dysregulation of miR-142 and miR-150 are reported in leukemia, gastric and lung cancers, but their roles in breast cancer or BCSCs are not elucidated. miR-142 is involved in hematopoiesis, immune responses, and T cell differentiation (*Chen and Lodish, 2005*; *Ramkissoon et al., 2006*; *Wu et al., 2007*; *Visone et al., 2009*) and is upregulated in bronchoalveolar stem cells (*Qian et al., 2008*), suggesting it might have a role in the regulation of tissue stem cells. miR-142 is upregulated in human T-cell acute lymphoblastic leukemia (*Lv et al., 2012*) but is downregulated in acute myeloid

leukemia (*Wang et al., 2012*). miR-150 is upregulated in adult T-cell leukemia cells (*Bellon et al., 2009*), gastric cancers (*Wu et al., 2010*), and lung cancers (*Zhang et al., 2013*).

The canonical Wnt pathway signal is transduced by β-catenin and plays a critical role in many adult stem cells, including those of the breast and intestine (*Reya and Clevers, 2005*; *Zeng and Nusse, 2010*). The fact that some cancer cells share the extended self-renewal ability with normal stem cells, and that the canonical Wnt signaling pathway is implicated in both stem cell self-renewal and cancer suggests that normal physiological regulators of stem cell functions might be 'hijacked' in cancer (*Reya and Clevers, 2005*). A variety of putative transcriptional targets of the canonical Wnt signaling pathway, such as c-Myc and cyclin D1, are identified (*He et al., 1998*; *Shtutman et al., 1999*). Adenomatous polyposis coli (APC) is a component of the destruction complex that destabilizes β-catenin and suppresses the activity of the canonical WNT signaling pathway. In human colon cancers, mutations in the *APC* gene are the most commonly known acquired genetic change for the aberrant activation of the canonical WNT signaling pathway during the tumor development and progression (*Kinzler et al., 1991*; *Nishisho et al., 1991*; *Cottrell et al., 1992*). In a model for the stepwise progression of the colon tumorigenesis, *APC* gene mutations play an important role in the initiation step followed by successive mutations in other genes, including K-Ras and p53 (*Fearon and Vogelstein, 1990*). During clonal progression, dysregulation of downstream β-catenin targets, such as c-Myc, can relieve colon cancer cells of their dependence on β-catenin signaling (*van de Wetering et al., 2002*). The expression of APC is not limited to the intestine but is widely observed in many other tissues, including lung, liver, kidney, and mammary tissue. However, the role of the suppression of APC and the activation of the canonical WNT signaling in the tumor initiation process of the tissues other than the colon largely remains unknown, because *APC* mutations are less frequent in tumors originating from these tissues. For example, recent data from the TCGA reveals an ~2% incidence of *APC* mutations in breast cancer (the TCGA Research Network: http://cancergenome.nih.gov/).

Dampening of APC inhibition of β-catenin appears to be more often due to promoter methylation (36–54%) and loss of heterozygosity (LOH) (23%) than to somatically acquired *APC* mutations (~2%) in human breast cancers (*Sarrio et al., 2003*; *Jin et al., 2001*; *Banerji et al., 2012*; *The cancer genome atlas network 2012*), but the presence of the *APC* promoter methylation and LOH are independent of tumor size or stage (*Jin et al., 2001*; *Sarrio et al., 2003*). The *APC* mutations in breast cancers are typically found at sites distinct from the *APC* mutation cluster region in colorectal cancers and are much more frequently seen in advanced than early stage breast cancers (*Furuuchi et al., 2000*). On the other hand, APC is targeted by the miRNAs, such as miR-27, miR-155, and miR-142 (*Lu et al., 2009*; *Wang and Xu, 2010*; *Liu et al., 2011*; *Hu et al., 2013*). These observations suggest that in addition to LOH, promoter methylation, and the *APC* mutations, miRNAs that target APC may regulate the aberrant activation of the canonical WNT signaling pathway for the initiation of human breast cancers, the enhancement of niche independence, and the aberrant proliferation of the human BCSCs.

In this study, we studied the roles of miR-142 and miR-150, two of the miRNAs that are upregulated in the human BCSCs, in the enhancement of the canonical WNT signaling pathway and in the regulation of human BCSCs. We employed the Ago IP/microarray method to show the relevance of the targeting of APC by miR-142, but not by miR-150, and identified various target mRNAs that were efficiently recruited to RISC by these miRNAs. Upregulation of miR-142 enhanced the canonical WNT signaling pathway and transactivated the expression of miR-150. Both microRNAs had the ability to induce hyperproliferation in mammary tissue. Finally, we show that knockdown of miR-142 reduced the clonogenicity of BCSCs in vitro and tumor growth in vivo. Our results provide the insights into the roles and mechanisms of two of the upregulated miRNAs in human BCSCs.

## Results

### miR-142 efficiently recruits the *APC* mRNA to a RISC

miRNAs suppress protein production by recruiting target mRNAs to the RISC where they associate with Ago, a core component of the RISC. To explore the possible target mRNAs of miR-142 and miR-150 which were more highly expressed in human BCSCs than in the NTG cells, we performed Ago IP/microarray to comprehensively identify mRNAs that were recruited to Ago by miR-142 and miR-150. Lysates from human embryonal kidney (HEK) 293T cells transfected with or without the precursor for miR-142 or miR-150 were immunopurified using an anti-Ago antibody. Amplified RNAs from the Ago-immunopurified samples were labeled and hybridized to the Human Exonic Evidence

Based Oligonucleotide (HEEBO) microarrays (*Klapholz-Brown et al., 2007*; *Hendrickson et al., 2008*). We identified dozens of potential targets of miR-142 and miR-150 (*Figure 1A* and *Figure 1A—source data 1–4*). The types of miRNA seed match (i.e. the association of target mRNA with the region centered on the nucleotides 2–7 of the 5′ region of miRNAs) correlate with the efficiency of mRNA targeting by miRNAs (*Bartel, 2009*). The efficiency of the miR142-dependent and miR-150-dependent recruitment of specific mRNAs to the RISC complex correlated well with the types of miRNA seed matches: mRNAs with the 8mer match were most efficiently enriched, followed by those with the 7mer-m8, 7mer-1A, and 6mer sites (*Figure 1B* and *Figure 1—figure supplement 1*). Although both miR-142 and miR-150 were predicted to target the *APC* mRNA by a computer algorithm, the results of the Ago IP/microarray experiments showed that the *APC* mRNA was efficiently recruited to RISC by miR-142, and that it was hardly recruited to RISC by miR-150 (*Figure 1A—source data 1–4*). Among the genes associated with the activity of the WNT signaling pathway, we found that the *APC* mRNA was strongly recruited to Ago by miR-142 (*Figure 1C*). In the parallel experiments, amplified RNAs from the whole cell lysate from HEK293T cells transfected with or without the miR-142 precursor were labeled and hybridized to the HEEBO microarrays. The results showed that the abundance of the *APC* mRNA was consistently reduced in the miR-142-transfected HEK293T cells (*Supplementary file 1*).

## miR-142 directly targets APC and activates the canonical WNT signaling pathway

We tested the ability of miR-142 to regulate the 3' UTR of the *APC* mRNA using a luciferase reporter assay. The 3′ UTR of the *APC* mRNA, which contained a predicted target site for miR-142-3p was cloned into the pGL3-MC vector downstream of a luciferase minigene (*Figure 2A*). HEK293T cells, which very weakly express miR-142, were co-transfected with the pGL3 luciferase-APC 3′UTR vector, or the control luciferase vector, pRL-TK Renilla luciferase vector, and miRNA precursors. We observed that when the 3′ UTR of the *APC* mRNA was included in the luciferase transcript, co-transfection of the miR-142 precursor suppressed the luciferase activity by 37% (*Figure 2B*). Mutations of the two nucleotides within the predicted target site for miR-142-3p in the 3′ UTR of the *APC* mRNA significantly weakened the ability of miR-142 to suppress the luciferase activity (*Figure 2A,B*), suggesting that miR-142 specifically targets the predicted seed sequence in the 3′ UTR of the *APC* mRNA to suppress the translation of the *APC* mRNA.

Then we tested the ability of miR-142 to regulate expression of endogenous APC protein. Western blot analyses showed that, after 6 days, the protein level of APC was decreased in the HEK293T cells expressing miR-142 when compared to the control precursor transfected cells (*Figure 2C*). To confirm our findings, breast cancer cells, such as MCF7 and MDA-MB-231 cells, were transfected with the miR-142 precursor. We found that the protein level of APC was reduced in both MCF7 and MDA-MB-231 cells (*Figure 2D*). Next, the expression of miR-142-3p was knocked down using an anti-miR-142-3p-expressing lentivirus in the HEK293T, MDA-MB-231, and MCF7 cells which expressed miR-142 at low levels. We found that the protein level of APC was elevated in HEK293T, MDA-MB-231, and MCF7 cells transduced with the anti-miR-142-3p vector (*Figure 2E* and data not shown). These results suggest that miR-142 effectively targets the *APC* mRNA and reduces the protein level of APC in the cells including the human breast cancer cells.

Inhibition of APC expression by miR-142 could, in principle, diminish the dependence of the canonical WNT signaling pathway on extrinsic WNT signals. To test this possibility, we evaluated the effects of miR-142 on the β-catenin-dependent transcription of the luciferase gene using cells transfected with a TOPFlash luciferase plasmid, which contained the β-catenin/TCF binding sites in the promoter region of a luciferase gene. As a control, cells were transfected with an otherwise identical FOPFlash plasmid, in which the β-catenin/TCF binding sites were mutated. The ratio of the measured luciferase activity of the TOPFlash plasmid to that of the FOPFlash plasmid provides a measure of the activity of the WNT/β-catenin signaling in the cells (*van de Wetering et al., 1997*). HEK293T cells were transfected with the miR-142 precursor and incubated with 20 mM lithium chloride (LiCl) for 6 hr to stimulate the canonical WNT signaling pathway (by inhibiting the activity of the glycogen synthase kinase 3) (*van Noort et al., 2002*). We found that the normalized TOPFlash/FOPFlash ratio of the luciferase activities increased by 364% in the miR-142 transfected HEK293T cells compared to that in the control precursor transfected cells (*Figure 3A*). In accordance with this result, the elevation of the normalized TOPFlash/FOPFlash value in the miR-142 transfected cells was observed by the stimulation with

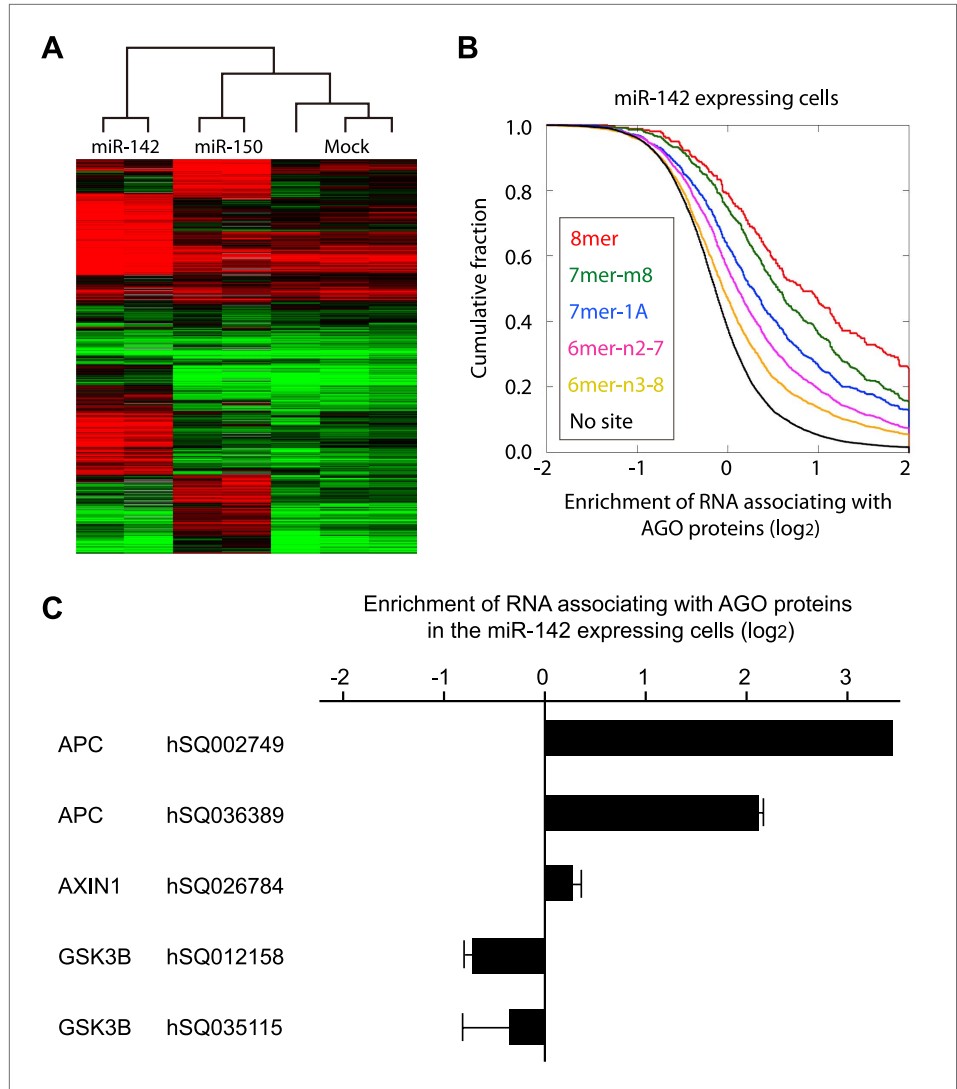

**Figure 1**. Recruitment of the *APC* mRNA to Ago by miR-142 or miR-150. (**A**) Unsupervised hierarchal cluster of AgoIP/microarray from HEK293T cells transfected with mock, miR-142- or miR-150-precursor. The lysates of HEK293T cells transfected with mock, or 30 nM miR-142- or miR-150- precursor were immunopurified by an anti-Ago antibody. RNA was isolated from the immunopurified lysate and amplified for the HEEBO microarray analyses. Rows correspond to the putative miR-142 and miR-150 targets (local false discovery rate (FDR) 1%), and columns represent individual experimental samples. (**B**) Cumulative distribution of the change for the Ago IP mRNAs classified by the types of the seed matches in the 3' UTR (**Bartel, 2009**). Overall efficiency of Ago IP enrichment is 8mer > 7mer-m8 > 7mer-1A > 6mer-n2-7 (nucleotides 2–7) > 6mer-n3-8 (nucleotides 3–8) in the HEK293T cells transfected with miR-142 precursor. (**C**) Ago IP enrichment of mRNAs. The mRNA enrichment in the Ago IP samples of the miR-142-expressing HEK293T cells over those of the mock transfected cells. The results are derived from two independent transfections. The results for APC, AXIN1, and GSK3B enrichment are presented. The data are mean ± standard deviation (SD).

The following source data and figure supplements are available for figure 1:

**Source data 1**.
**Source data 2**.
**Source data 3**.

*Figure 1. Continued on next page*

*Figure 1. Continued*

**Source data 4**.
**Figure supplement 1**.

Wnt3A (*Figure 3A*). To further confirm that miR-142 activated the canonical WNT signaling pathway by targeting APC, we co-transfected the APC expression plasmid, pCMV-Neo-APC, that contains the complete coding sequence and the partial 3′ UTR sequence of the *APC* mRNA, along with the TOPFlash or FOPFlash vectors and the miRNA precursors. The miR-142-3p target site within the pCMV-Neo-APC plasmid sequence was mutated to produce the pCMV-APC mutant plasmid (*Figure 3B*). The co-transfection of the control plasmid, pCMV-Neo-Control, or the pCMV-Neo-APC plasmid that contains miR-142-3p targeted sequence within the 3′ UTR region, did not affect the ability of miR-142 to elevate the normalized TOPFlash/FOPFlash value. On the contrary, co-transfection of pCMV-Neo-APC mutant plasmid significantly suppressed the normalized TOPFlash/FOPFlash value to an intensity comparable to that of the control miRNA precursor transfected cells (*Figure 3C*). These results suggest that the activation of canonical WNT signaling pathway by miR-142 is mostly mediated by the ability of miR-142 to reduce the protein level of APC.

## Inhibition of miR-142 suppresses the clonogenicity of BCSCs

The organoid culture system of the mammary tumor cells allows the in vitro formation of the organoids that maintain cellular hierarchy of the original tissue from which it derived (*Sato et al., 2009*; *Zeng and Nusse, 2010*). In this system, the tissue derived stem cells were embedded in Matrigel and cultured in the media that contained a cocktail of the growth factors, including ones that stimulate the WNT signaling pathways. We and others have previously shown that the MLTV-Wnt-1 tumor contains a CSC population (*Shackleton et al., 2006*; *Cho et al., 2008*). Because miR-142 is upregulated in the BCSCs and stimulates the canonical WNT signaling pathway that is an important regulator of the stem cell properties in mammary tissues, we tested the ability of miR-142 to regulate the organoid formation of BCSCs. The sorted murine mammary CSCs of the MLTV-Wnt-1 tumor were infected with the control lentivirus or the anti-miR-142-3p-expressing lentivirus that suppressed the expression of miR-142-3p and expressed GFP. The GFP-positive cells infected with the control lentivirus formed round-shaped organoids in 7 days (*Figure 4—figure supplement 1A*). In contrast, most of the cells infected with the anti-miR-142-3p-expressing lentiviruses failed to form the round-shaped organoids (*Figure 4—figure supplement 1A*). The number of the GFP positive organoids was significantly decreased when the mammary CSCs were infected with the anti-miR-142-3p-expressing lentivirus (*Figure 4—figure supplement 1B*). These results suggest that miR-142-3p is an important regulator of the organoid formation ability in murine mammary CSCs.

To confirm that these findings are applicable to the human BCSCs, we infected the human breast cancer cells with the anti-miR-142-3p-expressing lentiviruses and evaluated the ability to form the organoids derived from the human BCSCs. BCSCs of the human breast cancer xenograft tumors were infected with the control lentivirus or the anti-miR-142-3p-expressing lentiviruses. The GFP positive cells infected with the control lentivirus formed round-shaped organoids in 10 days (*Figure 4A*). In contrast, most of the cells infected with the anti-miR-142-3p-expressing lentiviruses failed to form the round-shaped organoids (*Figure 4A*). The number of the GFP positive organoids was significantly decreased when the human BCSCs were infected with the anti-miR-142-3p-expressing lentivirus (*Figure 4B*). Then we analyzed the effects of miR-142 on the cell proliferation and apoptosis of the cancer cells in the organoids. The organoids were incubated with 5-ethynyl-2′-deoxyuridine (EdU), and the percentage of the EdU positive cells was analyzed by flow cytometry. We found that the percentage of the EdU-positive cells in the GFP-positive organoids was significantly reduced when the human BCSCs were infected with the anti-miR-142-3p-expressing lentivirus (*Figure 4C*). Then we analyzed the percentage of apoptotic cells in the GFP-positive organoids by flow cytometry. We found that the percentage of the annexin V-positive cells in the GFP positive organoids was significantly increased when the human BCSCs were infected with the anti-miR-142-3p-expressing lentivirus (*Figure 4D*). These results suggest that miR-142 critically affects the clonogenic properties of the BCSCs by regulating the proliferation and apoptosis of the breast cancer cells.

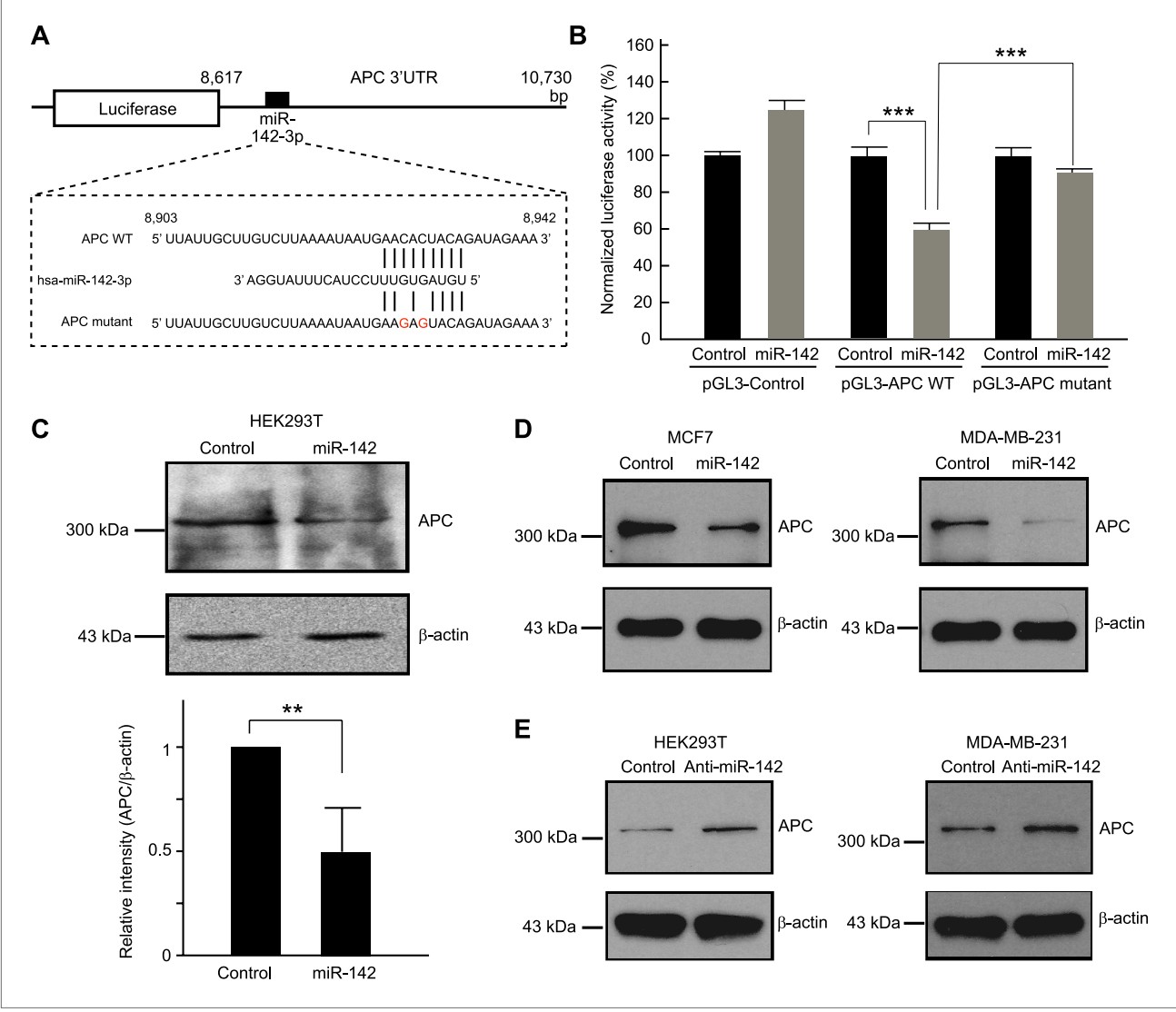

**Figure 2**. Targeting of APC by miR-142. (**A**) Schematic representation of the predicted miR-142 target site within the 3' UTR of *APC*. The predicted target site for miR-142-3p is located at the proximal portion of the *APC* 3' UTR. Two nucleotides complementary to the seed sequence (the nucleotides 2–7 of miRNA) of miR-142-3p were mutated in the APC mutant plasmid. The number indicates the position of the nucleotides in the reference wild-type sequence of *APC* (NM_000038.5). (**B**) Activity of luciferase gene linked to the 3' UTR of *APC*. The pGL3 luciferase reporter plasmids with the wild-type or mutated 3' UTR of *APC* were transiently transfected into HEK293T cells along with 25 nM miR-142 precursor or negative control precursor. Co-transfected Renilla luciferase reporter was used for normalization. Luciferase activities were measured after 48 hr. The mean of the results from the cells transfected by pGL3-Control vector with control precursor was set at 100%. The data are mean ± SD (n = 3, ***p < 0.005). (**C**) miR-142 suppressed endogenous APC expression. miR-142-expressing HEK293T cells were cultured for 6 days, and APC protein level was analyzed by Western blotting. The intensities of the bands for APC and β-actin were measured by ImageJ software. Difference in the APC protein level between the lysate of the control precursor trans-fected cells and that of miR-142 transfected cells was statistically significant (n = 4, **p < 0.01). (**D**) Suppression of endogenous APC by miR-142 in the breast cancer cells. miR-142-expressing breast cancer MCF7 and MDA-MB-231 cells were cultured for 2 days, and the APC protein level was analyzed by Western blotting. (**E**) Elevation of the endogenous APC protein level by miR-142 knockdown. HEK293T cells and MDA-MB-231 cells were infected with the anti-miR-142-3p-expressing lentivirus, and GFP-positive cells were sorted by a cell-sorter. APC protein level was analyzed by Western blotting.

## miR-142 upregulates the transcription of miR-150 through the canonical WNT signaling pathway

Both miR-142 and miR-150 are more highly expressed in BCSCs relative to the non-tumorigenic breast cancer cells (*Shimono et al., 2009*). Because the promoter region of miR-150 contains a potential β-catenin/TCF transcription factor binding site (*Figure 5A*), we investigated whether miR-150 expression

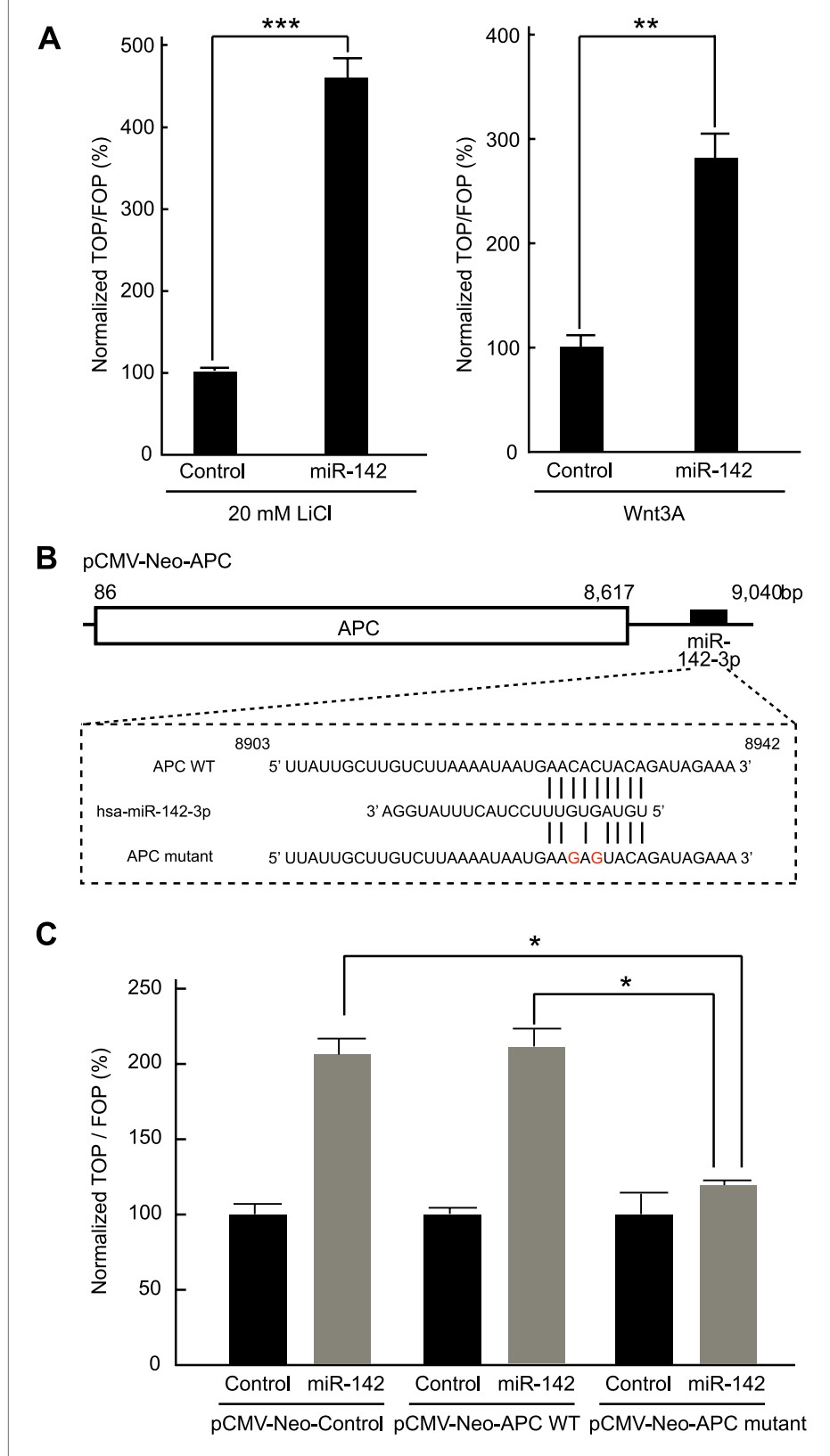

**Figure 3**. Activation of the canonical WNT signaling pathway by miR-142. (**A**) Activation of the canonical WNT signaling pathway by miR-142. HEK293T cells were transfected with 25 nM miR-142 precursor along with TOPFlash
*Figure 3. Continued on next page*

*Figure 3. Continued*

or FOPFlash vector (with or without TCF binding sites). Cells were stimulated with 20 mM LiCl (left) or with Wnt3A conditioned medium (right) for 6 hr. The canonical WNT signaling pathway activities were measured by dividing a normalized TOPFlash value by a normalized FOPFlash value. The mean of the results from the cells transfected with control precursor was set at 100%. The data are mean ± SD (n = 3, **p < 0.01, ***p < 0.005). (**B**) Schematic representation of the pCMV-Neo-APC expression vector that codes the full-length *APC* sequence together with wild-type or mutated miR-142 targeted site located at the 3′ UTR. Two nucleotides complementary to the seed sequence of miR-142-3p were mutated in the pCMV-Neo-APC mutant plasmid. The number indicates the position of the nucleotides in the reference wild-type sequence of *APC* (NM_000038.5). (**C**) miR-142 targets APC to activate the canonical WNT signaling pathway. HEK293T cells were transfected with 25 nM miR-142 precursor along with the TOPFlash or FOPFlash vector and the pCMV-Neo-APC expression vector or its mutant vector. Cells were stimulated with 20 mM LiCl for 6 hr. The activities of the canonical WNT signaling pathway were measured as described in (**A**). The mean of the results from the cells transfected with control precursor was set at 100%. The data are mean ± SD (n = 3, *p < 0.05).

is regulated by the canonical WNT signaling pathway. HEK293T cells were stimulated with 20 mM LiCl for 7 hr, then crosslinked with formaldehyde, and chromatin immunoprecipitation was performed either with an antibody against β-catenin or, as a control, with mouse IgG. The anti-β-catenin antibody selectively enriched DNA fragments containing the potential β-catenin/TCF binding site within the promoter of the miR-150 precursor (*Figure 5B*). Then, using semi-quantitative real-time PCR, we found that the expression of miR-142 was able to upregulate the expression of miR-150 in human breast cancer MDA-MB-231 cells (*Figure 5C*). Moreover, inhibition of the canonical WNT signaling pathway using the siRNA against β-catenin reduced the expression of miR-150 in the miR-142-expressing MDA-MB-231 cells (*Figure 5D*). We further confirmed that the inhibition of miR-142 in human breast cancer organoids formed by human BCSCs infected with the anti-miR-142-3p-expressing lentiviruses decreased the expression of miR-150 (*Figure 5E*). Taken together, these results indicate that the upregulation of miR-142 induces the expression of miR-150 at least partially through the canonical WNT signaling pathway.

## miR-142 induces mammary dysplasia and miR-150 induces hyperplasia

To clarify the effect of the upregulation of miR-142 and miR-150 on the mammary tissue in vivo, murine mammary epithelial cells were transduced with the lentiviruses expressing miR-142 or miR-150 and the regenerated mammary tissues were histologically analyzed. Expression of miR-142 or miR-150 in the lentivirus infected cells was confirmed by real-time PCR (data not shown). We infected $5 \times 10^4$ lineage⁻ murine mammary epithelial cells with the miR-142 or miR-150-expressing lentivirus and transplanted them into the cleared mammary fat pads of syngeneic mice. Non-infected and control lentivirus infected mammary cells were used as controls. Whole mount mammary tissue analyses and histological analyses were performed 8 weeks after transplantation. Transduction of the lentivirus into the mammary transplants was confirmed by the expression of ZsGreen. Overall, 7 out of 12 transplants with non-infected mammary cells and 7 out of 12 transplants with mammary cells infected with control lentivirus formed a mammary tree, suggesting that lentivirus infection was highly efficient and, by itself, did not perturb engraftment of mammary cells. Whole-mount staining and histological analysis of mammary fat pads injected with the control lentivirus-infected mammary cells showed the outgrowth of a normal-looking mammary tree structure consisting of single layers of myoepithelial and luminal cells (*Figure 6A,B*). In contrast, mammary cells infected with the miR-142 lentivirus formed disorganized structures with multiple layers of cells (*Figure 6A,B*). Histological analysis revealed that miR-142-expressing mammary cells formed clusters of cells with an abnormal appearance in which the normal mammary structure was disrupted because of the aberrant cell proliferation (*Figure 6B*). Mammary cells infected with the miR-150-expressing lentivirus formed a hyperplastic mammary tree with extremely increased branching and thick mammary ducts (*Figure 6A,C*). Histological analysis of the mammary trees showed that miR-150-expressing mammary cells formed thick hyperplastic ducts with multiple layers of mammary cells, with scattered small cavities among mammary cell layers (*Figure 6B,C*). But unlike miR-142-expressing mammary cells, miR-150-expressing mammary cells had a normal cellular appearance. To confirm that both the dysplastic mammary tissue formed by the miR-142-expressing mammary cells and the hyperplastic mammary tissue formed by

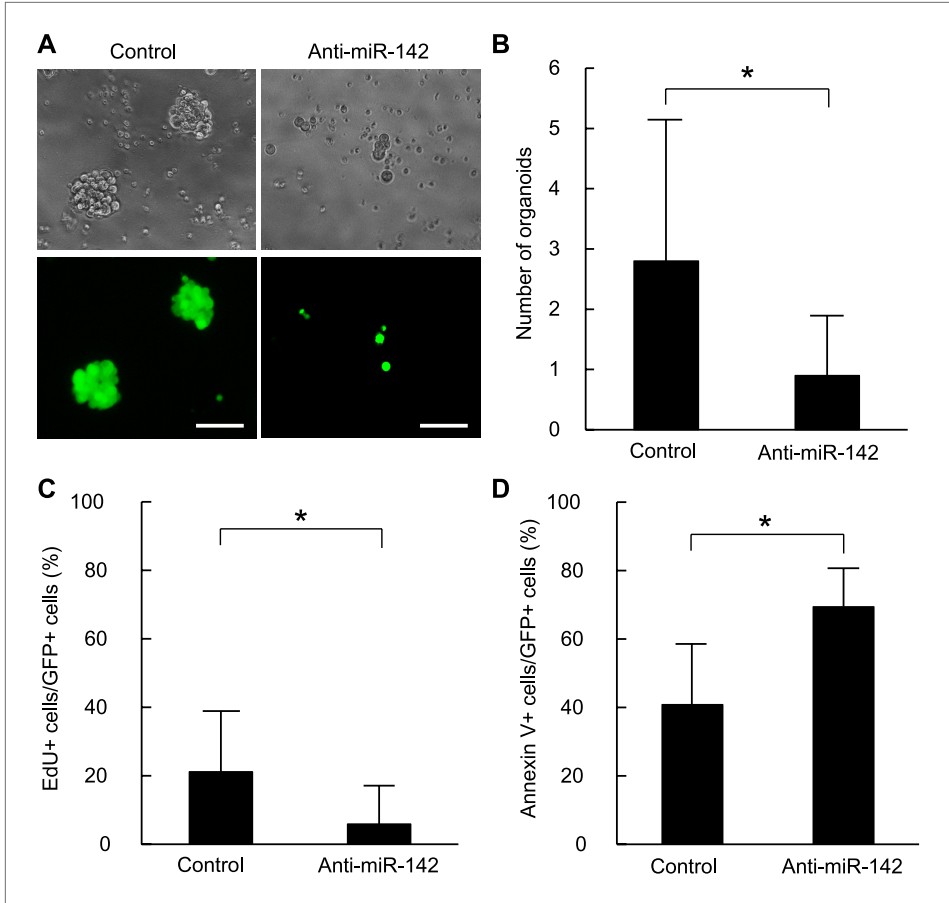

**Figure 4**. Suppression of the organid formation by the inhibition of miR-142 in BCSCs. (**A**) Representative images of the organoids from the human BCSCs infected with control or anti-miR-142-3p-expressing lentiviruses and cultured for 10 days. The upper panels are the phase-contrast images of the organoids, and the lower panels are the fluorescent microscopic images for the detection of GFP. Bars, 100 μm. (**B**) The number of organoids formed by the human BCSCs infected with control or anti-miR-142-3p-expressing lentiviruses. The data are mean ± SD (n = 10, *p < 0.05). (**C**) Percentage of the EdU-positive cells among the GFP-positive breast cancer cells in the organoids formed by the human BCSCs infected with control or anti-miR-142-3p-expressing lentiviruses. The data are mean ± SD (n = 4, *p < 0.05). (**D**) Percentage of the annexin V-positive cells among the GFP-positive breast cancer cells in the organoids formed by the human BCSCs infected with control or anti-miR-142-3p-expressing lentiviruses. The data are mean ± SD (n = 4, *p < 0.05).
The following figure supplement is available for figure 4:

**Figure supplement 1**.

the miR-150-expressing mammary cells were both hyperproliferative, we immunostained these tissues with an anti-PCNA antibody. We found that the epithelial cells in the miR-142-expressing and miR-150-expressing mammary tissues were positively stained with an anti-PCNA antibody (*Figure 6D*). In contrast, the mammary tissue regenerated by the control lentivirus transfected mammary cells was rarely stained with an anti-PCNA antibody. The results indicate that the enforced expression of either miR-142 or miR-150 induces hyperproliferation in the mammary tissues and that the phenotype of the miR-142-expressing mammary tissue is more severe and accompanied with abnormal morphology.

Next, we analyzed the expression and localization of β-catenin in the regenerated mammary tissues. While the signal for the β-catenin in mammary tissue formed by the control lentivirus-infected mammary cells was weakly detectable and its localization was mostly membranous, we observed cytoplasmic and nuclear localization of β-catenin and the positive staining of active β-catenin in the

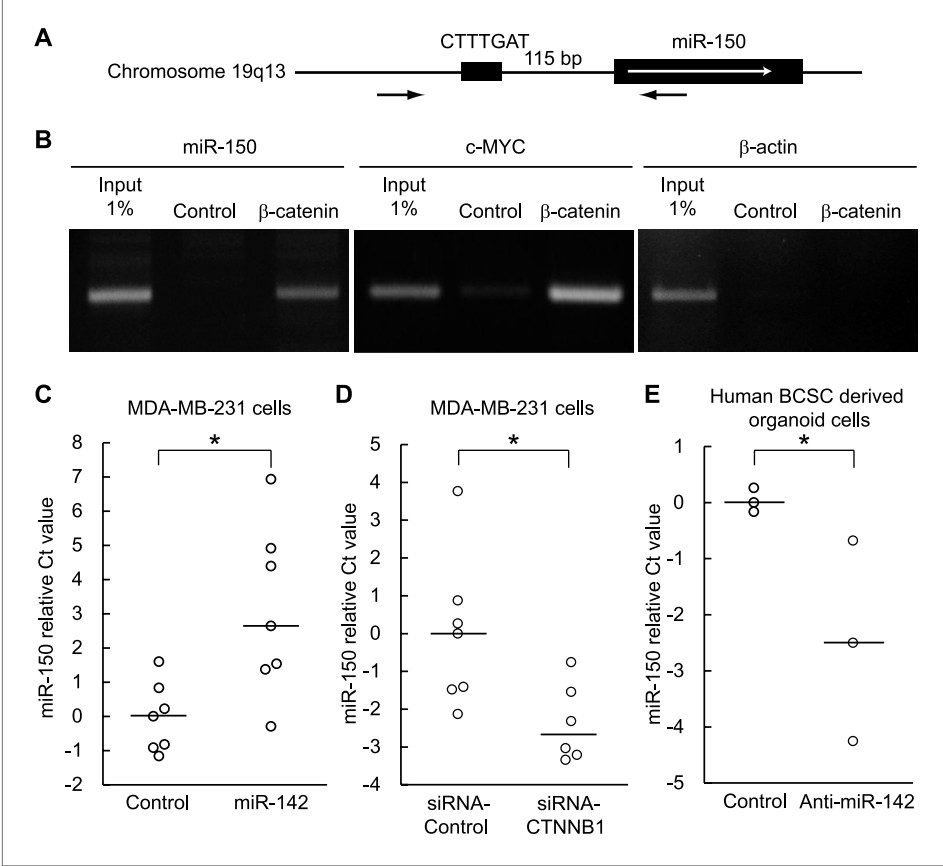

**Figure 5**. Enhancement of miR-150 transcription by the canonical WNT signaling pathway. (**A**) Schematic representation of the potential β-catenin/TCF binding site within the flanking genomic sequence of the miR-150 precursor. Box: potential β-catenin/TCF binding site (WWCAAWG/CWTTGWW) and its sequence. Box with an arrow: position of the miR-150 precursor and direction of transcription. Black arrows: relative positions of the PCR primers for the chromatin IP analyses. (**B**) Chromatin IP for a potential β-catenin/TCF binding site. Lysate of cross-linked HEK293T cells was immunoprecipitated by an anti-β-catenin antibody or mouse IgG. A putative β-catenin/TCF binding site was amplified by PCR. The template for input was purified from the 1% of total cell lysate. The β-catenin/TCF binding site for c-MYC was amplified as a positive control, and the genomic sequence for β-actin was amplified as a negative control. (**C**) miR-142 induced the transcription of miR-150. The amount of miR-150 in the MDA-MB-231 cells transfected with a miR-142-expressing plasmid was analyzed by quantitative real-time PCR. Each circle represents one experiment. Bars indicate median. Differences of the amount of miR-150 between the miR-142-expressing and the control MDA-MB-231 cells were statistically significant (*$p < 0.05$). (**D**) Decrease of the miR-150 transcription by β-catenin knockdown. β-catenin was knocked down by the siRNA against CTNNB1 in MDA-MB-231 cells expressing miR-142. Each circle represents one experiment. Bars indicate median. The amount of miR-150 was analyzed by quantitative real-time PCR. Difference in the amount of miR-150 between the cells transfected with a siRNA against CTNNB1 and those transfected with a control siRNA was statistically significant (*$p < 0.05$). (**E**) Decrease of miR-150 expression by the miR-142 inhibition in human BCSCs. The human BCSCs derived from the human breast cancer xenograft were infected with the anti-miR-142-expressing or control letiviruses and incubated in an organoid culture medium for 48 hr. The amount of miR-150 was analyzed by quantitative real-time PCR. Each circle represents one experiment. Bars indicate median (*$p < 0.05$).

dysplastic cell clusters formed by the mammary cells infected with the miR-142-expressing lentivirus (*Figure 6—figure supplement 1*). Although the immunohistochemistry of β-catenin in general suffers from non-specific staining, we detected nuclear β-catenin in the miR-142-expressing mammary tissue. We speculate that nuclear β-catenin was detectable in these immunohistochemistry experiments because the WNT signaling pathway was highly overstimulated in the miR-142-expressing mammary tissue. These results suggest that miR-142 regulates the properties of mammary cells by activating the WNT signaling pathway and conferring the aberrant proliferative ability.

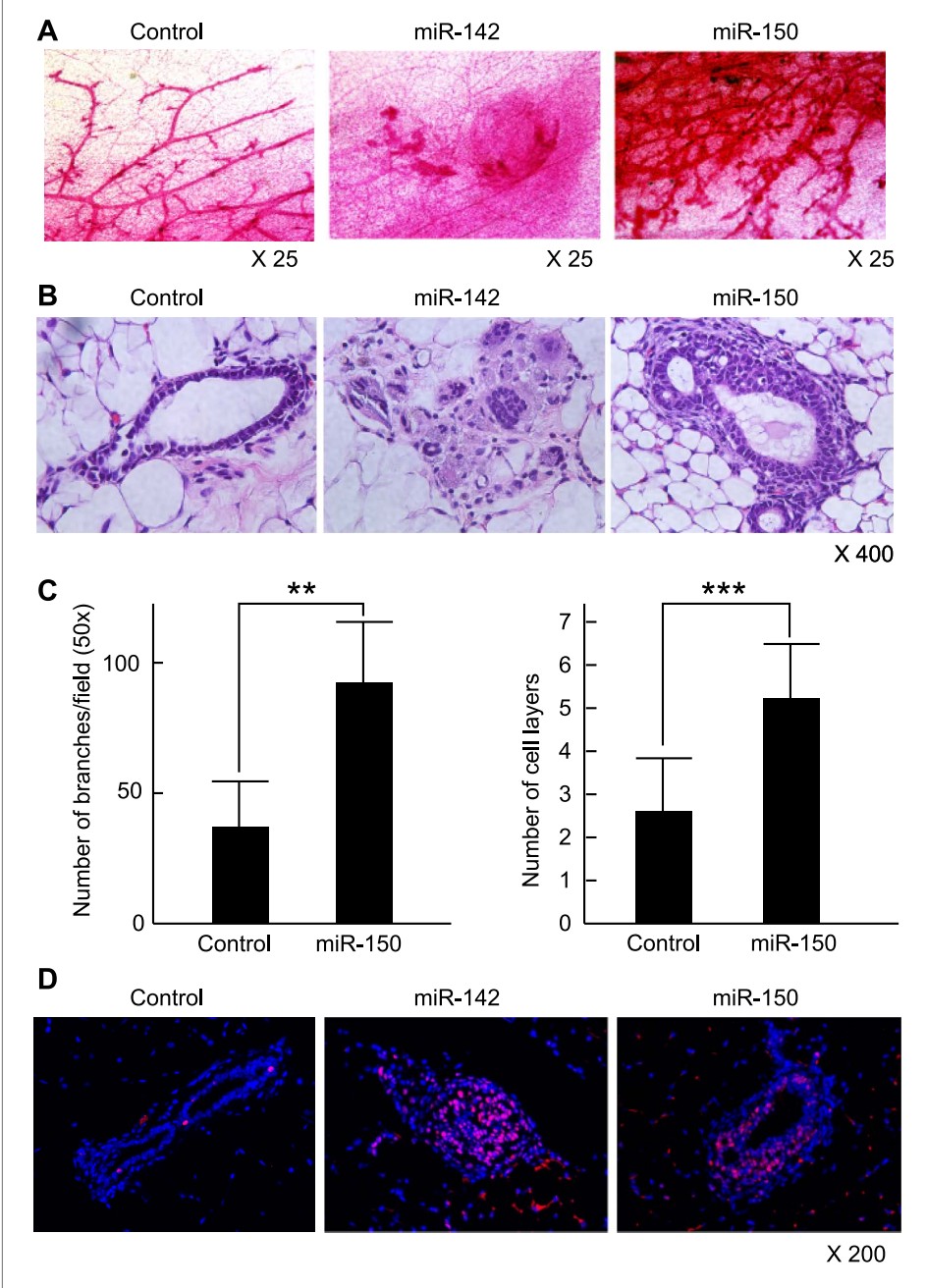

**Figure 6**. Mammary dysplasia or hyperplasia formed by the mammary cells expressing miR-142 or miR-150 in vivo. (**A**) The whole-mount mammary fat pad staining. Murine mammary cells isolated from FVB/NJ mice were infected with the control lentivirus, or miR-142 or miR-150-expressing lentiviruses. The $5 \times 10^4$ infected mammary cells were transplanted into cleared mammary fat pads of the same strain weaning age female mice. Mammary duct outgrowth was analyzed 8 weeks later. The whole-mount mammary tissue was stained by a carmine alum staining solution. (**B**) Hematoxylin and Eosin staining of the mammary tissue formed by mammary cells infected with miR-142 or miR-150-expressing lentiviruses. Mammary duct outgrowth was analyzed 8 weeks after transplantation. (**C**) Degree of branching and number of mammary epithelial cell layers in the mammary tissue formed by mammary cells infected with miR-150. Degree of branching was analyzed by counting the number of branches observed within a 50× field of the whole-mount mammary fat pad specimens. Number of mammary epithelial cell layers of the ducts was counted in the hematoxylin and eosin stained specimens (**p < 0.01, ***p < 0.005). (**D**) Increase of the cell proliferation in the mammary tissues regenerated by the miR-142- or miR-150-expressing mammary cells. *Figure 6. Continued on next page*

*Figure 6. Continued*

The tissues were stained with an anti-PCNA antibody followed by an Alexa Fluor 594-conjugated secondary antibody. Blue, DAPI; red, PCNA.

The following figure supplement is available for figure 6:

**Figure supplement 1**. Activation of β-catenin in the mammary tissue expressing miR-142.

## Inhibition of miR-142 suppresses the tumor growth initiated by human BCSCs in vivo

To evaluate the role of miR-142 in the growth of human breast cancers initiated by the human BCSCs, we infected human BCSCs isolated from a patient-derived human breast cancer xenograft (PDX) with the anti-miR-142-3p-expressing lentivirus or the control lentivirus. Then, $1 \times 10^4$ of the infected BCSCs were injected into the mammary fat pad region of the NSG mice. The flow cytometric analyses revealed that 94% and 96% of the human BCSCs were infected with the control or the anti-miR-142-3p-expressing lentivirus, respectively, 2 days after the infection (data not shown). The growth of the human breast cancer xenografts formed by the anti-miR-142-3p-expressing BCSCs was significantly slower than that of the control human breast cancer xenografts (*Figure 7A*). We further confirmed that consistent with the results of the human breast cancer cell lines (*Figure 2*), the protein level of APC was elevated in the human breast cancer cells isolated from the anti-miR-142-3p-expressing breast cancer xenograft (*Figure 7B*). These results suggest that the inhibition of miR-142-3p in human BCSCs from this patient elevated the expression of APC protein and suppressed the proliferation of the breast cancer cells in vivo.

## Discussion

Our results demonstrate that APC is indeed a relevant target of miR-142. In both normal and malignant mammary cells, miR-142 activates the canonical WNT signaling pathway and regulates proliferation at least in part through upregulation of the WNT/β-catenin signaling. However, because each miRNA can have hundreds of targets in general, it is likely that other pathways also have roles in the regulation of normal and malignant breast epithelium by miR-142.

Expression of APC protein is modulated by miRNAs, such as miR-135, miR-27, miR-155, and miR-142 (*Nagel et al., 2008*; *Lu et al., 2009*; *Wang and Xu, 2010*; *Hu et al., 2013*). Elevated expression of miR-135, which targets the *APC* mRNA, has been observed in colorectal adenomas and adenocarcinomas with/without biallelic *APC* mutations, suggesting that this miRNA could play a primary or synergistic role in activation of the canonical WNT signaling pathway during colon cancer development (*Nagel et al., 2008*). In a model of osteoblast differentiation, miR-27 and miR-142 were shown to regulate APC expression and to positively modulate the canonical WNT signaling pathway (*Wang and Xu, 2010*; *Hu et al., 2013*). miR-155 targets APC and inhibition of miR-155 using the anti-miR-155 increases the protein level of APC (*Lu et al., 2009*). It is noteworthy that both miR-142-3p and miR-155 are more highly expressed in the human BCSCs than in the NTG cells (*Shimono et al., 2009*). In this study, we show that miR-142 reduced the protein level of APC and enhanced the canonical WNT signaling pathway in normal and malignant mammary epithelium. This is important for the stem cells in the mammary gland (*Haegebarth and Clevers, 2009*; *Zeng and Nusse, 2010*). The dysregulated WNT signaling can enhance niche independence and promote aberrant proliferation of stem cells. Taken together, upregulation of miR-142, miR-150, and miR-155 may not only be hallmarks of BCSCs but could actually contribute directly to aberrant proliferation of these cells. In this report, we show that miR-150 is a target of miR-142 via its regulation of the canonical WNT/β-catenin signaling. Notably, miR-150 plays a role in the proliferation induced by miR-142 and the canonical WNT/β-catenin signaling. Our results suggest that miR-150's regulation of proliferation is downstream of the canonical WNT/β-catenin signaling. In addition, the *APC* mRNA has a predicted target site for miR-150. We constructed the pGL3 luciferase expression plasmid in which the miR-150 target site within the 3′ UTR of the *APC* mRNA was cloned downstream of a luciferase minigene and found that the activity of this luciferase plasmid was reduced by 32% when the miR-150 precursor was co-transfected (data not shown). However, the results of the Ago IP/microarray experiments revealed that *APC* mRNA was much less efficiently recruited to RISC by miR-150 (*Figure 1A—source data 3–4*). Because each

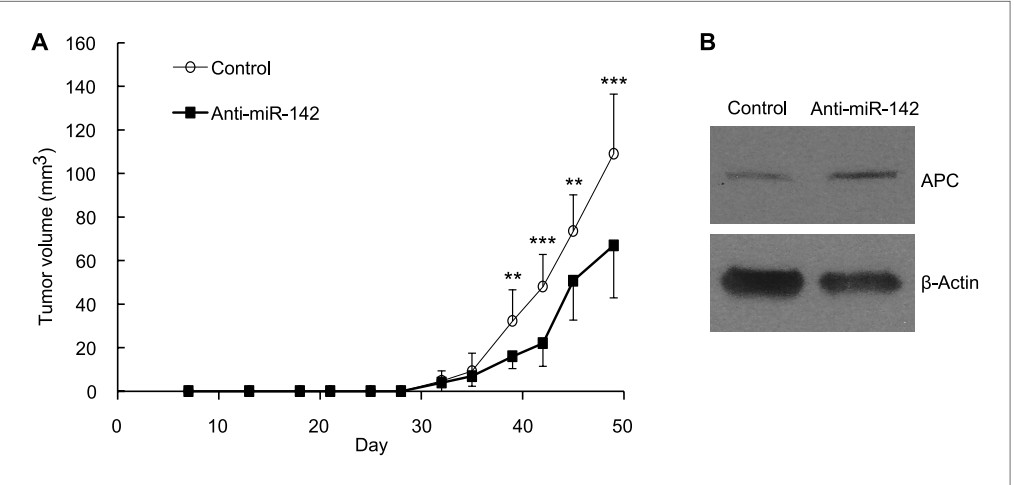

**Figure 7**. Suppression of the Tumor Growth Initiated by Human BCSCs Expressing the Anti-miR-142-3p in vivo. (**A**) CD44+ CD24-/low lineage− human BCSCs were isolated from an early passage human breast xenograft tumor and infected by the anti-miR-142-3p-expressing lentivirus or control lentivirus. Ten thousand infected cells were injected into the mammary fat pad region of immunodeficient NSG mice. Tumor growth was monitored for 2 months after injection. The data are mean ± SD (n = 10, * *p < 0.01, ***p < 0.005). (**B**) APC expression in the xenograft tumors was derived from the control or anti-miR-142-3p-expressing lentivirus infected BCSCs. Tumors were dissociated and APC expression was analyzed by western blotting.

miRNA has multiple target genes, we speculate that the presence of many other target genes with higher affinity to the miR-150-containing RISC will perturb the ability of miR-150 to regulate the *APC* mRNA and that miR-150 has much weaker effect, if any, on the WNT signaling pathway in vivo.

Some miRNAs are known to be regulated by transcriptional factors which have critical roles in regulating development, proliferation, and cell survival (*Bracken et al., 2008*; *O'Donnell et al., 2005*; *Raver-Shapira et al., 2007*; *Yamakuchi et al., 2010*; *Chang et al., 2007*; *Ma et al., 2007*). For example, the transcriptional repressors, ZEB1 and ZEB2, which can play a positive role in EMT, suppress transcription of the miR-200a-miR-200b-miR-429 cluster, which in turn represses expression of ZEB1 and ZEB2, thereby forming a double negative feedback loop (*Bracken et al., 2008*). c-Myc enhances expression of the miR-17-92 cluster, which in turn represses E2F1 expression and thereby suppresses cell-cycle entry (*O'Donnell et al., 2005*). The tumor suppressor Tp53 regulates expression of miRNAs, including miR-34a, miR-107, and miR-200 (*Chang et al., 2007*; *Raver-Shapira et al., 2007*; *Yamakuchi et al., 2010*; *Chang et al., 2011*). Expression of miR-10b, a miRNA that promotes breast cancer cell metastasis by inhibiting HoxD10, is regulated by the transcription factor Twist, a major regulator of the epithelial-to-mesenchymal transition (*Ma et al., 2007*). We identified a TCF/β-catenin binding site in the promoter region of miR-150 and showed that miR-142 stimulates the canonical WNT signaling pathway in vivo and elevates the expression of miR-150. It has recently been reported that miR-150 induces proliferation and suppresses apoptosis of breast cancer cells (*Huang et al., 2013*). A simple model to explain the upregulation of miR-142 and miR-150 in human BCSCs is that suppression of the APC protein expression by miR-142 increases the activity of the canonical WNT signaling pathway and thereby enhances the miR-150 expression. We note that our results show that miR-142 upregulates the expression of miR-150 when miR-142 is highly expressed and/or the WNT signaling pathway is strongly stimulated in mammary epithelial cells. Considering that miR-142 is highly expressed in human BCSCs but weakly expressed or undetectable in the stem/progenitor population of the mammary epithelial cells, our result suggests that the upregulation of miR-142 and the increase of the miR-150 expression by miR-142 could be especially relevant in the initiation and/or progression of human breast cancers driven by human BCSCs in vivo.

The results of the mouse models suggest that APC dysregulation is associated with initiation and progression of at least some breast cancers. Mice heterozygous for a germline mutation in *Apc* (Apc+/850T, ApcMin mice) spontaneously develop mammary tumors, although at significantly lower frequency than intestinal tumors (*Moser et al., 1993*). Mice heterozygous for a C-terminal truncation mutation,

Apc[+/1572T], develop multifocal mammary tumors with pulmonary metastasis at much higher frequency (*Gaspar et al., 2009*). MMTV-promoter-Wnt1 transgenic mice develop breast tumors composed of both luminal and myoepithelial component and expanded mammary stem cell pools (*Cho et al., 2008*). Finally, heterozygous mutation of *APC* in immature, but not mature, mammary epithelium induces breast tumors (*Kuraguchi et al., 2009*). These results suggest that activation of the canonical WNT signaling pathway in mammary epithelium, by mutation of *APC*, excessive ectopic WNT expression, and/or stabilization of β-catenin contribute to mammary tumor initiation in the same way as colon cancer at least in mouse models of the mammary tumors. However, although these mouse models that upregulate the canonical WNT signaling activity induce mammary gland tumors (*Moser et al., 1993*; *Gaspar et al., 2009*; *Kuraguchi et al., 2009*), *APC* mutations are less frequently found in human breast cancers (*Furuuchi et al., 2000*; *Jin et al., 2001*; *Sarrio et al., 2003*). APC is a key tumor suppressor that regulates the canonical WNT signaling pathway and is involved in development and homeostasis of a variety of cells including stem cells (*Reya and Clevers, 2005*). In addition, APC has additional important cellular functions, including roles in cell adhesion, migration, organization of actin and microtubule networks, spindle formation, and chromosome segregation (*Aoki and Taketo, 2007*). Thus, dysregulation of these processes by mutations in the *APC* gene is frequently implicated in tumor initiation and progression. Our results show that upregulation of miR-142 can significantly enhance the canonical WNT signaling pathway through the suppression of APC including in the mammary cells. Therefore, miR-142, a miRNA frequently upregulated in human BCSCs than in the NTG cells (*Shimono et al., 2009*), could provide at least a part of the shared molecular mechanism for aberrant activation of the canonical WNT signaling pathway in BCSCs and for the initiation and progression of breast cancers.

## Materials and methods

### Ethics statement

Primary breast cancer specimens and normal breast tissues were obtained from the consented patients as approved by the Research Ethics Boards at Stanford University and at the City of Hope Cancer Center in California. All animal experiments were carried out under the approval of the Administrative Panel on Laboratory Animal Care of Stanford University.

### Cell culture

Human embryonic kidney (HEK) 293T cells, and MCF7, and MDA-MB-231 breast carcinoma cells were maintained in Dulbecco's modified Eagle's medium (DMEM) with 10% FBS, 100 U/ml penicillin, 100 µg/ml streptomycin with or without 250 ng/ml amphotericin B (Invitrogen, Carlsbad, CA) and incubated at 5% $CO_2$ at 37°C.

### RNA isolation and Ago IP

HEK293T cells were seeded at $6 \times 10^6$ cells in 10-cm tissue culture plates 12 hr prior to transfection. All transfections were carried out with Lipofectamine 2000 (Invitrogen), according to the manufacturer's instructions. A set of four plates was transfected with same miRNA precursors and each plate was transfected with 300 pmol miRNA precursor (Ambion, Austin, TX). Plates were washed with 1× ice-cold phosphate-buffered saline (PBS) at 48 hr post-transfection and 700 µl of an ice-cold lysis buffer (150 mM KCl, 25 mM Tris–HCl, pH 7.4, 5 mM EDTA, 0.5% Nonidet P-40, 0.5 mM DTT, 100 U/ml SUPERase In (Ambion) with Complete proteinase inhibitor (Roche, Germany)) was added to a 10-cm plate. After incubating for 30 min at 4°C, plates were scraped and the lysates were combined and spun at 14,000 rpm at 4°C for 30 min. The supernatant was collected and filtered through a 0.45 µm syringe filter. The biotinylated anti-Argonaute antibody (12.5 µg; biotin was conjugated to an anti-human Ago antibody (#01-2203, Wako, Japan)) (*Hendrickson et al., 2009*) was mixed with 250 µl Dynal M-280 Streptavidin coated magnetic beads (Invitrogen), which were equilibrated by washing three times with lysis buffer and twice with PBS. The beads were incubated with the lysate at 4°C for 4 hr and washed twice with 10× volume of lysis buffer for 5 min. Five percent of the beads were frozen for SDS-PAGE analysis after the second wash. RNA was extracted directly from the remaining beads with 25:24:1 phenol:chloroform:isoamyl alcohol (Invitrogen). Trace amounts of phenol were removed by chloroform extraction and RNA was precipitated using Glycogen (Invitrogen) as a carrier. RNA pellets were resuspended in 30 µl of RNase free water and stored at −80°C.

## Microarray production and pre-hybridization processing

Detailed methods for microarray experiments are available at the Brown lab website (http://cmgm.stanford.edu/pbrown/protocols/index.html). HEEBO oligonucleotide microarrays were produced by Stanford Functional Genomic Facility. The HEEBO microarrays contain ~45,000 70-mer oligonucleotide probes, representing ~30,000 unique genes. A detailed description of this probe set can be found at http://microarray.org/sfgf/heebo.do. RNA from IP experiments was hybridized to microarrays printed on epoxysilane glass (*Hendrickson et al., 2009*).

## Sample preparation, hybridization, and washing

For HEEBO microarray experiments, poly-adenylated RNAs were amplified in the presence of aminoallyl-UTP with Amino Allyl MessageAmp II aRNA kit (Ambion). For expression experiments, universal reference RNA was used as an internal standard to enable reliable comparison of relative transcript levels in multiple samples (Stratagene, La Jolla, CA). Amplified RNA (3–10 µg) was fluorescently labeled with NHS-monoester Cy5 or Cy3 (GE Healthcare, United Kingdom). Dye-labeled RNA was fragmented, then diluted in a 50 µl solution containing 3× SSC, 25 mM HEPES-NaOH, pH 7.0, 20 µg human Cot-1 DNA (Invitrogen), 20 µg poly(A) RNA (Sigma-Aldrich, St.Louis, MO), 25 µg yeast tRNA (Invitrogen), and 0.3% SDS. The sample was incubated at 70°C for 5 min, spun at 14,000 rpm for 10 min in a microcentrifuge, then hybridized at 65°C for 12–16 hr (*Hendrickson et al., 2009*). Following hybridization, microarrays were washed in a series of four solutions containing 400 ml of 2× SSC with 0.05% SDS, 2× SSC, 1× SSC, and 0.2× SSC, respectively. The first wash was performed at 65°C for 5 min. The subsequent washes were performed at room temperatures for 2 min each. Following the last wash, the microarrays were dried by centrifugation in a low-ozone environment (<5 ppb) to prevent the destruction of Cy dyes (*Fare et al., 2003*). Once dry, the microarrays were kept in a low-ozone environment during storage and scanning (see http://cmgm.stanford.edu/pbrown/protocols/index.html).

## Scanning and data processing

Microarrays were scanned using either AxonScanner 4200 or 4000B (Molecular Devices, Sunnyvale, CA). PMT levels were auto-adjusted to achieve 0.1–0.25% pixel saturation. Each element was located and analyzed using GenePix Pro 5.0 (Molecular Devices). Data were filtered to exclude elements that did not have a regression correlation of ≥0.6 between Cy5 and Cy3 signal over the pixels compromising the array element of and intensity/background ratio of ≥2.5 in at least one channel, for 60% of the arrays. For cluster and SAM analysis of Ago+/− miRNA IPs vs mock IPs, measurements corresponding to oligonucleotides that map to the same EntrezID were treated separately and the data were globally normalized per array, such that the median $\log_2$ ratio was 0 after normalization.

## Plasmid vectors and mutagenesis

A 525 bp fragment of the *APC* 3′ UTR (corresponding to the positions of 8589–9113 of the NM_000038.5) was amplified by PCR using the cDNA of HEK293T cells as a template and cloned into the pGEM-T-Easy vector. Then the *APC* 3′ UTR product was cloned at the 3′ of the luciferase gene of the pGL3-MC vector (*Shimono et al., 2009*). All products were sequenced. The pCMV-Neo-Bam APC plasmid was obtained from Addgene (Cambridge, MA). To produce the control pCMV-Neo-Control plasmid, the pCMV-Neo-Bam-APC plasmid was digested by BamHI and ligated. Mutation of the putative miR-142-3p target sequences within the 3′ UTR of *APC* and within the pCMV-Neo-Bam-APC plasmid was generated using the QuikChange Site-Directed Mutagenesis kit (Stratagene).

## Lentivirus backbone plasmids

The sequences of hsa-miR-142 including stem loop structure and 200–300 base pairs of up-stream and down-stream flanking regions were cloned by PCR using genomic DNA of HEK293 cells as a template. The products were cloned into multicloning sites of pEIZ-HIV-ZsGreen vector (kind gift from Dr. Zena Werb, UCSF) (*Welm et al., 2008*). Lentiviruses were produced as described (*Tiscornia et al., 2006*). For knockdown of miR-142-3p, the miRNA Zip (anti-miRNA) plasmid specific for miR-142-3p (MZIP142-3p-PA-1) together with the scrambled control RNA-expressing plasmid was purchased from System Bioscience (Mountain View, CA) and lentiviruses were produced, according to the manufacturer's instructions.

## Luciferase assay and TOPFlash/FOPFlash reporter assay

HEK293T cells were seeded at $1 \times 10^5$ cells per well in 48-well plates the day prior to transfection. All transfections were carried out with Lipofectamine 2000 (Invitrogen), according to the manufacturer's instructions. Cells were transfected with 320 ng pGL3 luciferase expression construct containing the 3′ UTR of human *APC*, 40 ng pGL4.74 hRLuc/TK Renilla luciferase vector (Promega, Fitchburg, WI), and 25 nM hsa-miR-142 precursor or negative control precursor (Ambion). Forty-eight hr after transfection, luciferase activities were measured using the Dual-Luciferase Reporter Assay System (Promega) and normalized to Renilla luciferase activity. For the TOPFlash/FOPFlash Reporter assays, cells were transfected with 320 ng TOPFlash or FOPFlash luciferase expression construct containing TCF/β-catenin binding sites, 40 ng pRL-TK Renilla luciferase vector (Promega), and 25 nM hsa-miR-142 precursor or negative control precursor (Ambion). To co-transfect pCMV-Neo-APC plasmids, cells were transfected with 200 ng TOPFlash or FOPFlash luciferase expression construct, 20 ng pRL-TK Renilla luciferase vector (Promega), 300 ng pCMV-Neo-APC plasmids, and 25 nM hsa-miR-142 precursor or negative control precursor (Ambion). 48 h after transfection, cells were stimulated by 20 mM LiCl or 15% Wnt3A conditioned medium for 6 hr. Luciferase activities were measured using the Dual-Luciferase Reporter Assay System (Promega) and normalized to Renilla luciferase activity. All experiments were performed in triplicate.

## Western blotting

The cells were seeded at $7 \times 10^5$ cells in a 6-well plate, transfected with miR-142-precursor, and cultured for 2 days. To analyze the effect of anti-miR-142-3p in breast cancer cells and human breast cancer xenograft cells, the cells were infected with the anti-miR-142-3p-expressing lentivirus and the GFP-expressing cells were collected using a cell sorter. Cells were washed with PBS twice and soaked in hypotonic buffer (10 mM Tris–HCl pH 7.5, 5 mM $MgCl_2$) with Complete proteinase inhibitor (Roche) on ice for 20 min and sonicated. Then an SDS sample buffer (50 mM Tris–HCl, pH 6.8, 2% SDS, 10% glycerol 5 mM EDTA, 0.02% bromophenol blue, 3% β-mercaptoethanol) was added to the total cell lysate. Samples were separated on SDS-4-15% gradient polyacrylamide gel electrophoresis and transferred to polyvinylidene difluoride filters (Amersham). After blocking with 5% skim milk in 0.05% Tween 20/PBS, filters were incubated with 1:100 diluted anti-APC polyclonal antibody (C-20, sc-896, Santa Cruz Biotechnology, Dallas, TX) or 1:500 diluted anti-β-actin monoclonal antibody (C4, sc-47778, Santa Cruz Biotechnology). Then 1:5000-10,000 diluted peroxidase-conjugated sheep anti-rabbit or mouse IgG antibody (Amersham, United Kingdom) was added and developed using the SuperSignal West Dura Substrate (Thermo Scientific, Waltham, MA). The experiments were repeated four times and the intensity of the band was measured using the ImageJ software.

## Organoid-formation assay

Mouse MMTV-Wnt1 breast tumors were digested and CD45-Thy1+CD24+ murine breast cancer cells were isolated using a cell sorter as described previously (*Cho et al., 2008*). Three thousand CD45-Thy1+CD24+ murine breast cancer cells were sorted directly into 150 µl of the organoid medium (Advanced DMEM/F12 medium (Invitrogen) supplemented with 2 mM GlutaMax (Invitrogen), 10 mM HEPES (Sigma-Aldrich), 1 mM sodium pyruvate (Lonza, Switzerland), 10% FBS, 500 ng/ml R-spondin1 (R&D systems, Minneapolis, MN), 100 ng/ml Noggin (Peprotech, Rocky Hill, NJ), ITES media supplement (Lonza), 50 ng/ml EGF (Peprotech), 10 µM Y-27632 (Calbiochem, Gibbstown, NJ), 100 U/ml penicillin, 100 µg/ml streptomycin, 250 ng/ml amphotericin B (Invitrogen)). The cells were infected with 25 multiplicity of infection (moi) of control or anti-miR-142-3p-expressing lentiviruses and cultured in the ultra-low attachment 96-well plate for 12 hr. To prepare a feeder layer, $2 \times 10^3$ irradiated 3T3-L1 cells (Sigma-Aldrich) were seeded in a 96-well plate and cultured with a feeder medium (Advanced DMEM/F12 medium supplemented with 2 mM GlutaMax, 10 mM HEPES, 1 mM sodium pyruvate, 10% FBS, 100 U/ml penicillin, 100 µg/ml streptomycin, and 250 ng/ml amphotericin B (Invitrogen)) at 5% $CO_2$ at 37°C for 16 hr. Then feeder media were replaced by 40 µl of growth factor reduced Matrigel (BD Bioscience, Franklin Lakes, NJ). Three thousand lentivirus-infected MMTV-Wnt1 tumor cells in 150 µl of the organoid media were plated on the Matrigel and cultured at 5% $CO_2$ at 37°C.

The CD44+ CD24−/low human BCSCs were collected from the dissociated single cell suspension of the early-passaged human breast xenograft (COH69). The cells were infected with 20 moi of control or anti-miR-142-3p-expressing lentiviruses and cultured in the ultra-low attachment 96-well plate for 12 hr. To prepare a feeder layer, $1 \times 10^4$ irradiated L Wnt-3A cells (ATCC, Manassas, VA) were seeded

in a 96-well plate and cultured with a feeder medium at 5% $CO_2$ at 37°C for 16 hr. Then feeder media were replaced by 50 µl of growth factor reduced Matrigel (BD Bioscience). Five thousand lentivirus-infected human BCSCs in 100 µl of the organoid media were plated on the Matrigel and cultured at 5% $CO_2$ at 37°C. The organoids were observed using a microscope (Leica DMI 6000 B, Leica, Germany). The organoid cells were incubated with 10 µM EdU for 2 hr and the proliferation of the dissociated organoid cells was analyzed using the Click-iT Plus EdU Alexa Flour 647 Flow Cytometry Assay Kit (Life Technologies, Carlsbad, CA), according to the manufacturer's instructions. The flow cytometric analyses of apoptotic cells in the organoids were performed using an APC-conjugated Annexin V (BioLegend, San Diego, CA).

## Chromatin IP

Chromatin IP was performed using the Magna ChIP G kit (Upstate, Lake Placid, NY), according to the manufacturer's protocol. HEK293T cells were seeded at $2 \times 10^7$ cells in 15-cm tissue culture plates and cells were treated with 20 mM LiCl for 7 hr. Cells were crosslinked by adding formaldehyde directly to culture medium to a final concentration of 1% and incubated for 10 min, followed by incubation with glycine for 5 min. After washing twice with ice-cold PBS, cells were scraped and collected in a micro-centrifuge tube. Cell were precipitated by spinning at 3000 rpm at 4°C for 5 min and incubated with the lysis buffer with proteinase inhibitor on ice for 15 min. Tubes were spun, the nuclear lysis buffer with proteinase inhibitor was added and sonicated to form shared chromatin of about 200–1000 base pairs in length (data not shown). The product was diluted 10 times by ChIP dilution buffer, and 2 µg of an anti-β-catenin antibody (clone 14, BD Bioscience) or control normal mouse IgG (Santa Cruz Biotechnology) together with 20 µl of fully suspended protein G magnetic beads were added. Lysates were incubated overnight with rotation at 4°C and washed with low salt buffer, high salt buffer, LiCl buffer, and TE buffer. DNA was purified by using the column provided in the kit. The primer set was designed for the human genomic sequence flanking the putative TCF/β-catenin binding site (*Figure 5A*). The sequences of primers are: miR-150 forward, GTGTGCAGTTTCTGCGACTCAG; reverse, CACTGGTACAAGGGTTGGGAGAC; c-MYC forward, GCACGGAAGTAATACTCCTCTCCTC; reverse, CAGAAGAGACAAATCCCCTTTGCGC; β-actin forward, GTGTCTAAGACAGTGTTGTGGGT GTAG; reverse, CTGGGGTGTTGAAGGTCTCAAACATG. The amount of TCF/β-catenin biding sequence enriched by chromatin IP was evaluated by PCR and subsequent agarose gel electrophoresis.

## Real-time PCR assay

RT, pre-PCR, and the real-time PCR for miRNA expression analyses were performed by the real-time PCR method as described previously (*Tang et al., 2006*; *Shimono et al., 2009*). The abundance of each miRNA was measured individually by using the 7900HT Fast Real-Time PCR System (Applied Biosystems, Foster City, CA). Results were normalized by the amount of small nuclear RNA expression, C/D box 96A, C/D box84, or U6 snRNA.

Cells were transfected with miR-142-precursor and cultured for 2 days. To analyze the effect of anti-miR-142-3p in human breast cancer xenograft cells, the cells were infected with the anti-miR-142-3p-expressing lentivirus and the GFP-expressing cells were collected using a cell sorter. To knock-down β-catenin, the cells were transfected with siRNA against CTNNB1 (Life Technologies) or negative control siRNA (Life Technologies) using the Lipofectamine RNAiMAX reagent (Life Technologies). The cells were lysed with TRIzol (Life Technologies), and total RNAs were extracted following the manufacture's protocol. SuperScript VILO (Life Technologies) and TaqMan MicroRNA Reverse Transcription kit (Life Technologiies) were used for the mRNA and miRNA measurement experiments, respectively. Then, RT products were amplified with TaqMan PreAmp master Mix (Life Technologies), when necessary. The abundance of each mRNA or miRNA was measured using a 7900HT Fast Real-Time PCR system (Applied Biosystems).

## Mammary cell transplantation and whole-mount staining

Mammary epithelial cells were obtained as described previously (*Stingl et al., 2006*). Briefly, mammary glands from 8- to 14-week old virgin female FVB were digested with collagenase/hyaluronidase (StemCell Technologies, Canada). Single cell suspension was obtained by dissociation of the fragments with 0.25% trypsin, and dispase and DNase I (StemCell Technologies). Lineage⁻ mammary epithelial cells were obtained after removing the CD45+, Ter119+, and CD31+ cells using the EasySep mouse epithelial cell enrichment kit (StemCell Technologies). The isolated cells were mixed with 5 moi of lentivirus and incubated for 16 hr at 5% $CO_2$ at 37°C. Our flow cytometry analysis using the

lentivirus-infected mammary cells revealed that ~50% of the mammary cells were infected 1 day after infection. Fifty-thousand lentivirus infected cells were injected into cleared mammary fat pad of weaning age FVB/NJ female mouse. After 8 weeks, ZsGreen expression of the transplanted mammary tissue was checked under the fluorescent microscope (Leica DMI 6000 B). Whole-mount mammary tissue was fixed in Carnoy's fixative (60% ethanol, 30% chloroform, 10% glacial acetic acid) for 4 hr, and whole-mount staining was performed using carmine alum staining solution. For hematoxylin and eosin staining, the mammary tissue with ZsGreen expression was fixed in formalin, embedded in paraffin, and stained by hematoxylin and eosin staining method. The stained tissue was observed using a microscope (Leica DM 4000 B).

## Immunohistochemistory

Formalin-fixed paraffin-embedded tissue was cut in serial sections and mounted on glass slides. The tissue sections were deparaffinized and rehydrated. Antigen retrieval was performed using a 0.01 M citrate buffer (pH 6.0) by heating the sample in a microwave for 20 min. The slides were incubated with an anti-PCNA antibody (clone D3H8P, Cell Signaling, Danvers, MA), an anti-β-catenin antibody (1:200, clone 14, BD Bioscience), or an anti-active-β-catenin antibody (1:100, clone 8E7, Millipore, Billerica, MA) at 4°C overnight. Then tissue was stained by an Alexa Fluor594-conjugated secondary antibody (1:200, Jackson Laboratory, Bar Harbor, ME) or an Alexa Fluor 488-conjugated secondary antibody (1:200, Invitrogen) at room temperature for 1 hr. The stained tissue was observed using a fluorescent microscope (Leica DM 4000 B).

## Human breast cancer xenograft assay

The CD44[+] CD24[-/low] lineage[−] human BCSCs were isolated by a cell sorter. BCSCs were infected by 20 moi of anti-miR-142-3p-expressing lentivirus or control lentivirus by spin infection for 2 hr. Infected cells were washed with PBS and were mixed with Matrigel (BD Biosciences). Ten thousand infected cells were injected into mammary fat pad of female NSG mouse. Tumors were measured twice a week, and their volume was estimated using the formula: volume = $ab^2/2$ (a, length; b, width) (*Tanaka et al., 1990*). All experiments were carried out under the approval of the Administrative Panel on Laboratory Animal Care of Stanford University.

## Statistical analysis

Data from experiments were statically analyzed using T-test. For the results of real-time PCR, we employed Mann–Whitney U-test.

## Acknowledgements

This work was supported by the California Breast Cancer Research Program of the University of California #12FB-0053, and the Grants-in-Aid from the Japan Society for the Promotion of Science to YS, by the Fundacion Alfonso Martin Escudero and the Fulbright to MZ, by the Research Fellowship of the Japan Society for the Promotion of Science for Young Scientists to SH, by the Research Grant of the Research Foundation for Community Medicine, Japan, to TI, by the grants from the NIH (NIH CA100225, CA104987, CA126524, CA139490), the U.S. Department of Defense (W81XWH-11-1-0287, W81XWH-13-1-0281), the Breast Cancer Research Foundation, and the Ludwig Foundation to MFC, and by the NIH S10 Shared Instrumentation Grant #1S10RR02933801. The funders had no role in study design, data collection and analysis, decision to publish, or preparation of the manuscript.

## Additional information

### Competing interests

KL: This author is a principal scientist of the Applied Biosystems. MFC: Michael Clarke holds stock of the Oncomed Pharmaceuticals that focuses on development of therapeutic methods to target cancer stem cells. The other authors declare that no competing interests exist.

### Funding

| Funder | Grant reference number | Author |
| --- | --- | --- |
| California Breast Cancer Research Program | 12FB-0053 | Yohei Shimono |

| Funder | Grant reference number | Author |
|---|---|---|
| Japan Society for the Promotion of Science | 23130510, 25130707 | Yohei Shimono |
| Fundación Alfonso Martín Escudero | | Maider Zabala |
| Fulbright Commission | | Maider Zabala |
| Research Foundation for Community Medicine, Japan | | Taichi Isobe |
| National Institutes of Health | NIH CA100225, CA104987, CA126524, CA139490 | Michael F Clarke |
| Breast Cancer Research Foundation | | Michael F Clarke |
| Morton Family Foundation | | Michael F Clarke |
| Virginia and D.K. Ludwig Fund for Cancer Research | | Michael F Clarke |
| Japan Society for the Promotion of Science | | Shigeo Hisamori |
| U.S. Department of Defense | W81XWH-11-1-0287, W81XWH-13-1-0281 | Michael F Clarke |

The funders had no role in study design, data collection and interpretation, or the decision to submit the work for publication.

## Author contributions

TI, YS, Conception and design, Acquisition of data, Analysis and interpretation of data, Drafting or revising the article; DJH, Conception and design, Acquisition of data, Analysis and interpretation of data; SH, MZ, DGH, SC, AHK, SSS, JSL, Acquisition of data, Analysis and interpretation of data; PD, Acquisition of data, Analysis and interpretation of data, Contributed unpublished essential data or reagents; FS, Conception and design, Acquisition of data; DQ, Acquisition of data, Drafting or revising the article; FMD, GS, Acquisition of data, Contributed unpublished essential data or reagents; KL, Analysis and interpretation of data, Contributed unpublished essential data or reagents; POB, Conception and design, Analysis and interpretation of data; MFC, Conception and design, Analysis and interpretation of data, Drafting or revising the article

## Ethics

Human subjects: Primary breast cancer specimens and/or normal breast tissue were obtained from patients with informed consent in compliance to Protocol 4344, pre-approved by Stanford Institutional Review Board (IRB). Specifically, IRB Protocol 4344 authorizes acquisition of human breast tissue from the patients recruited in Stanford University Hospital and Clinics and City of Hope National Medical Center (COH), Los Angeles, then documented and delivered by Stanford Tissue Bank. Protocol 4344 waives the researchers in this study of patient consent requirement because (1) all patients recruited are pre-consented by physicians or surgeons in the Stanford Hospital and Clinics or in the COH under their IRB approval; (2) the researchers of this study do not have any direct contact with the recruited patients, (3) the researchers of this study do not receive any protected health information (PHI) of recruited patients; (4) all specimens received by the researches of this study are de-identified and labeled only as a sequential number, without any clinical association.

Animal experimentation: All animal experiments were carried out in compliance to Protocol 10868, pre-approved by Stanford University Administrative Panel on Laboratory Animal Care (APLAC). All researchers, who performed procedures using live animal, were pre-approved by Stanford APLAC, based on their completion of required animal use and care training, and acceptable previous experience in animal experiments.

## Additional files

### Supplementary file

• Supplementary file 1. Changes in mRNA abundance in the miR-142-expressing cells.

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
