## [Decision Letter]

Thank you for sending your work entitled “miR-142 regulates the tumorigenicity of human breast cancer stem cells through the canonical WNT signaling pathway” for consideration at *eLife*. Your article has been evaluated by a Senior editor and 4 reviewers, one of whom is a member of our Board of Reviewing Editors.

The Reviewing editor and the other reviewers discussed their comments before we reached this decision, and the Reviewing editor has assembled the following comments that we wish you to consider in deciding how to proceed.

All four reviewers find your model interesting and the work of potential interest to the broad readership of *eLife*. While reviewers 2 and 3 point out that there are studies on miR-150 that have previously linked it to APC and breast cancer tumor growth, your study investigating the functional role of two up-regulated miRNAs, miR-142 and miR-150, in breast cancer development, is intriguing and would represent a significant advance if substantiated further. Indeed the novelty rests upon showing that miR-142 over-expression regulates miR-150 expression, while miR-150 over-expression enhances canonical WNT signaling, and that this is physiologically relevant to tumor progression *in vivo*. The manuscript does provide interesting insights into the roles and mechanisms of miRNAs in the mammary gland. The major concern is that despite the attractiveness of your model, the major conclusions are not yet substantiated with sufficient evidence to make a compelling case.

Several relatively straightforward experiments could be performed that if successful would strengthen the work and make it more appropriate for *eLife*.

Major points:

1) Reviewer 2's suggestions for testing your model more rigorously seems reasonable, namely 1) to knockdown miR-142 in breast cancer cells (which you've already done) or mammary epithelial cells and perform qPCR for APC and miR-150. The prediction would be APC upregulation and miR-150 down-regulation. And 2) to inhibit WNT signaling in breast cancer cells or mammary epithelial cells and test whether this leads to reduced miR-150 expression.

2) Both reviewers 1 and 2 were troubled by the apparent discrepancy between the *in vivo* and *in vitro* data. Based on the model, miR-142 over-expression or inhibition should phenocopy miR-150 over-expression or inhibition. However, this doesn't seem to be the case as during mammary gland regeneration *in vivo*, miR-142 and miR-150 have opposite effects: one inhibits regeneration while the other promotes it even though in both cases, beta-catenin is stabilized and activated. It would be essential to reconcile this *in vivo* observation with the proposed mechanism derived from your *in vitro* study that both sit in the same pathway with miR-142 activating Wnt and miR-150 suppressing APC.

3) In regards to point 2 above, reviewers 1 and 3 point out that there is no demonstration that the breast cells overexpressing miR-142 eventually develop tumours or are more prone to getting tumours compared to wildtype cells. *In vivo* evidence is mandated to demonstrate that miR-142-induced dysplasia and miR-150-induced hyperplasia is relevant to actual breast cancer and not merely a correlation of expression level seems reasonable and critical to the paper. Reviewer 3 also touches on this issue, pointing out that the paper is about breast cancer stem cells, and since the effects on tumorigenicity are implied throughout the paper, it is important to demonstrate conversely that the miR-142 inhibited breast cancer cells display reduced tumorigenic capacity. Reviewers 2 and 4 echo the lack of rigorous testing of your *in vitro* data *in vivo* (for instance the efficacy and miRNA dosage of your shRNA-based strategy to knock down miR-142 and transfected miRNA precursors to overexpress miR-142), and point out the need to address whether WNT signaling is altered in the presence of miR-142 *in vivo*.

In addition, the reviewers raise a number of specific points that need to be shored up if this work were to be published in *eLife*. Many of them deal with seemingly missing controls and/or simple experiments to more rigorously challenge the *in vitro* model. Although reviewer 2's suggested rescue experiment and the need to repeat the Ago experiments in a more appropriate cell type could be excluded if the authors are able to address the above 3 key points, the other specific points seem within the authors' grasp of what could be carried out in a reasonable length of time.

At this point we request a written response that outlines how you could handle the requested revisions within a reasonable period of time. Our concern is how long it may take to perform the essential *in vivo* validation of the conclusions based on the *in vitro* experiments. The Board member will evaluate your responses and come to a decision concerning the appropriateness of your work for *eLife*.

*Reviewer #1*:

This manuscript investigates the functional role of two up-regulated miRNAs, miR-142 and miR-150, in breast cancer development. The authors propose an interesting idea, namely that miR-142 promotes breast cancer by directly targeting APC and thus activating Wnt-mediated miR-150 transactivation. Despite the attractiveness of their model, the authors do not support their major conclusions with sufficient evidence. The discrepancy between *in vitro* and *in vivo* observations further raises the doubt on the physiological relevance of miR-142/miR-150 function and underlying Wnt signaling, and this would need to be resolved prior to moving forward.

Indeed in this reviewer's assessment, the main concern is the disparity of *in vivo* and *in vitro* effects from miR-142 or miR-150 overexpression. During mammary gland regeneration *in vivo*, miR-142 and miR-150 have opposite effects: one inhibits regeneration while the other promotes it even though in both cases, beta-catenin is stabilized and activated. How do the authors reconcile this *in vivo* observation with the proposed mechanism derived from their *in vitro* study that both sit in the same pathway with miR-142 activating Wnt and miR-150 suppressing APC? It would seem important to provide *in vivo* evidence to demonstrate that miR-142-induced dysplasia and miR-150-induced hyperplasia is relevant to actual breast cancer; as it stands this is merely a correlation of expression level.

There are a few additional issues that seem important to address. It was not clear from the study how miR-142 and miR150 levels in cancer cells (BCSC or NTG) compare to levels in normal mammary tissue. Are these microRNAs up-regulated in tumor versus normal mammary tissue? The current data are not sufficient to conclude that the miR-142/APC/Wnt/miR-150 pathway is specifically required by the tumor-propagating cancer cells; it could just be a mechanism to bulk tumor growth.

Finally, what is the percentage of human breast cancers with APC suppression that are not due to somatic mutation, promoter methylation or LOH? This information would help to assess whether this miRNA-mediated APC inhibition is a relevant mechanism in human cancer.

Below, I delineate the experimental evidence that would need to be bolstered to substantiate the conclusions drawn:

Figure 1: What is the efficiency of the miR150-dependent recruitment of specific mRNAs to the RISC complex? Does this correlate with the types of miRNA seed matches as miR-142 does? The authors should minimally show control data that the miR-150 Ago2IP was efficient. To substantiate that miR-150 does not target APC as suggested by AgoIP, the authors should use luciferase assays to confirm.

Figure 2: What is the relative miR-142 overexpression level by the pre-miR-142 treatment in MCF7 and MDA-MB-231 cells? How much knock down is achieved with miR-142-3p miRZIP? These are important measurements to show how much endogenous miR-142-3p was perturbed by the various tools the authors used.

*Reviewer #2*:

In their manuscript entitled “miR-142 regulates the tumorigenicity of human breast cancer stem cells through the canonical WNT signaling pathway”, Shimono, Clarke, and colleagues present an interesting set of experiments that implicate miR-142 as a microRNA that is over-expressed in prospectively isolated tumorigenic human breast cancer populations. They present evidence, through Ago immunoprecipitation studies that APC mRNA is enriched in RISC upon miR-142 transfection. They show that miR-142 modulation regulates APC expression. They also show that miR-142 over-expression regulates miR-150 expression, while miR-150 over-expression enhances canonical WNT signaling. Finally, they show that miR-142 inhibition reduces breast cancer organoid formation, while miR-142 and miR-150 over-expression alters mammary gland organization.

This is an interesting report that implicates two microRNAs in the regulation of mammary gland development, breast cancer organoid formation, and WNT pathway regulation. The authors had previously profiled tumorigenic and non-tumorigenic human breast cancer populations and profiled their microRNA content. In their previous Cell publication, they identified miR-200c (a repressed microRNA) as a suppressor of tumor formation and mammary duct formation. In this work, they have identified 2 genes that are over-expressed in tumorigenic populations and have tested their roles in these phenotypes. Their conclusions on a positive role for miR-150 are supported by a recent paper by Huang et al., on a positive role for miR-150 in increasing breast cancer tumor growth (Huang et al., PLoS One, 2013). As I gather the overall pathway described consists of the following:

miR-142 –| APC→ +WNT signaling→ miR-150→ increase tumorigenicity/hyperplastic mammary gland phenotype

Major points:

1) If the above pathway is the model presented, a simple experiment would be to knockdown miR-142 in breast cancer cells (which the authors have already done) or mammary epithelial cells and perform qPCR for APC and miR-150. Prediction would be APC upregulation and miR-150 down-regulation. Have the authors done this?

2) A second prediction would be that inhibiting WNT signaling in breast cancer cells or mammary epithelial cells would lead to reduced miR-150 expression. This should be tested.

3) One thing that I don't understand is that based on the above model, miR-142 over-expression or inhibition should phenocopy miR-150 over-expression or inhibition. However, this doesn't seem to be the case as Figure 6 reveals opposite effects for miR-142 and miR-150. How do the authors reconcile this?

4) Have the authors tried to rescue the miR-142 organoid phenotype by introducing the APC coding sequence and determining if the effect is reversed?

5) Since the paper is about breast cancer stem cells and effects on tumorigenicity are implied throughout the paper, it is important for the authors to demonstrate that the miR-142 inhibited breast cancer cells display reduced tumorigenic capacity.

6) The authors should include control experiments on proliferation and apoptosis experiments upon miR-142 or miR-150 over-expression. I presume that there shouldn't be an effect since the effect is on renewal?

*Reviewer #3*:

In the present work, Shimono and colleagues study the effect of miR-142 and miR-150 to Wnt signaling *in vitro*, how their deletion affects the spheroid formation ability of mammary cancer stem cells in culture, and how their overexpression influences mammary gland development in mammary reconstitution assays. They claim that miR-142 and miR-150 induce Wnt signaling and that this results to deregulation of breast cancer stem cells (BCSC) and overproliferation of normal mammary SC. The role of miR-142 in inhibiting APC expression in HEK293T cells have been previously reported by Liu et al, Biochemical and Biophysical Research Communications 408 (2011) 259-264 and other groups cited by the authors. Thus, the novelty of this study stems from the discovery that this regulation of miR-142 on WNT signaling is also relevant in the mammary gland.

The topic of this study is interesting. However, there are 2 major weaknesses of the paper that needs to be addressed; firstly, the *in vitro* data presented was not subsequently tested rigorously *in vivo*, namely to address the question of whether WNT signaling is altered in the presence of miR-142. Secondly, while the central thesis of the authors pertains to the enhancement of tumorigenicity of BCSCs by miR-142, there is no demonstration that the breast cells overexpressing miR142 eventually develop tumours or are more prone to getting tumours compared to wildtype cells.

Overall, the claims of the paper were not well substantiated by the data presented. The authors should revise their conclusions to more clearly reflect the findings of the study. However, the manuscript does provide interesting insights into the roles and mechanisms of miRNAs in the mammary gland that is generally not well described in the literature. Thus the paper in its current state doesn't fit the quality and novelty requirements and we recommend that the manuscript should potentially be revised and reconsidered for publication in the future if the authors can satisfactory address the different comments raised.

Specific comments:

1) In Figure 2, the authors use sh-RNAs against miR-142 in HEK293T and MDA-MB-231 cells, and they observe downregulation of APC protein levels. As previously described and cited by the authors herein, the MDA-MB-231 and HEK293T cells express low levels of miR-142. How anti-miR-142 can affect the APC levels in these cell lines if they don't express this miRNA? The authors should use another cell line, expressing miR-142 at a significant level, in order to prove that it is indeed the specific activity of sh-miR-142 that is responsible for the downregulation of APC. Showing the levels of miR-142 before and after the use of shRNA is also essential.

2) With reference to Figure 2, there is a lack of mechanistic insight into how miR-142 inhibits organoid formation in BCSCs. While the authors have demonstrated that miR-142 stimulates canonical WNT signaling via the inhibition of APC in HEK293T, MCF7 and MDA-MB231 cell lines, the role of miR-142 in the regulation of WNT signaling in the CSC population of MMTV-Wnt1 tumours was not formally tested. What is the level of the APC protein and TOP/FOPflash readout in sh-miR-142 infected cells?

3) In Figure 4 the authors present the effect of anti-miR-142 to spheroid formation to BCSC from mouse and human mammary tumours. The authors miss a control here, which is same culture setting with the addition of a Wnt inhibitor (ShRNA against b-catenin).

4) The authors state that their results suggest that miR-142 and miR-150 regulate Wnt signaling in BCSCs. However, the only relevant data in the present manuscript is the spheroid-forming assay using shRNA against miR-142, without estimating the Wnt pathway activity with or without the addition of shRNA and without testing the effect of overexpression or inactivation of miR-150 in BCSC. They should perform spheroid forming assays with overexpression/deletion of both miRNAs, compare and combine these assays with Wnt pathway inhibition, and finally perform *in vivo* tumour forming assays.

5) From the results presented in Figure 5, the authors conclude in the corresponding text that miR-142 upregulates miR-150 via the canonical Wnt signaling. To prove this point, the authors should use miR-142 expressing cells and show that the levels of miR-150 decrease upon addition of shRNA against β-catenin.

6) More importantly, the Figure 6 shows that when mammary epithelial cells are transfected with miR-150 and grafted in empty mammary fat pads, they recapitulate the phenotype of activated Wnt signaling (largely hyperplastic mammary gland). However, when transfected with miR-142, they observe the opposite phenotype (complete destruction of the mammary ducts), which clearly does not phenocopy the previously reported phenotypes of Wnt gain of function in the mammary gland. If miR-142 activates both Wnt signaling and miR-150 expression, then miR-142 transfection should give a comparable phenotype. These results therefore contradict the model derived from the *in vitro* data, and indicate that maybe this is not the case in mammary epithelial SC. Could miR-142 be regulating another key player that may lead to the observed dysplasia? Can the phenotype in miR-142 transduced and grafted cells be rescued with the knockdown of miR-150? The authors should also test whether WNT signaling is changed in this context by crossing with WNT-signaling reporter mice (for example TOPGAL or bat-gal).

7) The experiments presented in Figure 6 should be repeated using the CSC population in MMTV-Wnt-1 tumours and human BCSCs in order to substantiate the claims of the authors made in the title of the manuscript.

8) The β-catenin staining presented in Figure 7 is not convincing. Immunofluorescence of β-catenin is known to give very high background and false positive results, and the assay of choice to reliably test the expression of nuclear b-catenin is immunohistochemistry with proper controls.

*Reviewer #4*:

In this manuscript, Shimono et al. characterized miR-142-3p's function in human breast cancer stem cells. The authors present evidence that when overexpressed in 293T cells miR-142 is capable of recruiting APC mRNA to the RISC. They further validated the direct regulation of APC by miR-142 by the luciferase reporter assays and protein measurement. Because APC negatively regulates the canonical WNT signaling, the authors demonstrated, by the reporter assays, that high expression of miR-142 stimulates the WNT pathway. In cultured cells, knocking down miR-142 significantly reduced the organoid formation capacity of BCSCs. Conversely, overexpression of miR-142 causes mammary dysplasia in a transplantation assay. Overall, this study provides an interesting link for the elevated miR-142 (and miR-150 to a lesser extent) expression observed in BCSCs to the activation of the canonical WNT signaling. However, several issues existing in this manuscript have dampened my enthusiasm to this work.

1) The regulation by microRNAs is highly sensitive to cellular context. It was not clear why the authors performed their elegant Ago/IP experiments in HEK293T cells with transfected miR-142. In this experiment, both mRNA context and the expression of miR-142 were artificial (Figure 1). The result would yield little insights for how miR-142 would function in BCSCs. Although the authors subsequently validated the targeting of APC by miR-142 in MCF7, MDA-MB-231 cells, these cells also don't present BCSCs as identified by the authors (Figure 2). Thus, it remains a major task to confirm that APC is indeed regulated by miR-142 in BCSCs and determine the level of regulation in this context. In addition, it is now widely accepted that an miRNA can target hundreds of mRNAs. It is unclear to me why APC stands out as a target of particular importance, other than its well-documented role in the WNT signaling. It is not sufficiently explored whether the regulation of APC is a major or minor function of miR-142. As such, it remains an open possibility that miR-142 regulates other genes also involved in the WNT activation or other completely different pathways. Without these studies, it remains unclear whether miR-142 indeed functions through its control of the WNT signaling.

2) The authors used an shRNA-based strategy to knock down miR-142 and transfected miR precursors to overexpress miR-142. The function of miRNAs is highly sensitive to its dosage (Mukherji S et al., 2011). Thus, it is critical to document the efficacy and miRNA dosage with such experimental manipulations. One could argue that the overexpression experiment introduces a very high level of miR-142 and generates artifacts due to the high levels. The authors, at minimum, should compare the levels of overexpressed miR-142 to the elevated miR-142 observed in BCSCs. On the other hand, the authors should also document the knock down efficiency by the means of Northern blotting, qPCR or reporter assays esp. for a knock-down method that is not very well documented.

3) In Figure 3, the authors performed an interesting experiment to overexpress APC with WT and the mutated miR-142 target site in the presence of miR-142. However, the 3'UTR used was not the full length but a fragment. It is demonstrated that the position of miRNA target sites could affect the strength of regulation (Grimson A et al., 2007). Thus, by using a shortened 3'UTR, the authors could potentially alter the 3'UTR context of the miR-142 target site that, in turn, leads to artifacts. They should use the full-length 3'UTRs in all assays.

4) In Figure 4, the authors performed a series of experiments to document to the role of miR-142 in mouse and human BCSCs. However, the characterization is superficial and limited to simple organoid formation assay. Is it caused by reduced proliferation and/or enhanced cell death and/or cell senescence? Is the repressed WNT signaling by miR-142 knock down responsible for such defects or other pathways were also involved? More careful analysis is warranted for this study.

5) In Figure 6, the authors carried out mammary gland regeneration assay to probe the functions of miR-142/150. However, it is not explained whether miR-142/150 are up-regulated in normal mammary gland or during mammary gland regeneration or during tumor initiation of breast cancer. Without such information, the defective mammary gland regeneration caused by overexpression of miR-142/150 provides little insights for the roles of these miRNAs in BCSCs.

[Editors’ note: further clarifications were requested prior to acceptance, as outlined below.]

Thank you for sending an outline of the proposed revision to the work entitled “miR-142 regulates the tumorigenicity of human breast cancer stem cells through the canonical WNT signaling pathway”.

For the most part, the experiments you plan to conduct to address point 1 are reasonable and acceptable.

However, there are several additional issues regarding point 2 that must be satisfactorily addressed if this work is to be seriously considered for publication in *eLife*:

1) The reviewers asked the authors to understand the discrepancy between the *in vivo* and *in vitro* data. Based on the model, miR-142 over-expression or inhibition should phenocopy miR-150 over-expression or inhibition. However, this doesn't seem to be the case as during mammary gland regeneration *in vivo*, miR-142 and miR-150 have opposite effects: one inhibits regeneration while the other promotes it even though in both cases, beta-catenin is stabilized and activated. It would be essential to reconcile this *in vivo* observation with the proposed mechanism derived from the *in vitro* study that both sit in the same pathway with miR-142 activating Wnt and miR-150 suppressing APC. The authors respond to this comment by saying that different micro-RNAs may regulate many different functions, which is probably true, but it does not help to understand their data and proposed mechanisms if they are not substantiated by their functional experiments *in vivo*.

2) The revised version will need a better explanation of how the authors reconcile the discordant phenotypes of miR-142 over-expression/inhibition with miR-150 over-expression/inhibition given that they are proposing these miRNAs are in a pathway.

3) It is well recognized that a single miRNA can target hundreds of targets, and miR-142 has been shown to target APC previously. The authors' central hypothesis is that miR-142's effect in breast cancer is mediated through the activation of the Wnt pathway that activates miR-150 as a result. This discrepancy should be experimentally addressed in the revised version.

---

## [Author Response]

Our new findings are summarized below:

1) The results of the inhibition of miR-142-3p in the human breast cancer xenograft cells and the Wnt signaling pathway inhibition in human breast cancer cells have confirmed that our finding that miR-142 targets APC, activates the Wnt signaling pathway and upregulates miR-150 is applicable to human breast cancers.

2) The results of the immunohistological experiments have confirmed that both the ‘dysplastic’ mammary tissues regenerated by the miR-142-expressing mammary cells and the ‘hyperproliferative’ mammary tissue regenerated by the miR-150-expressing mammary cells represent a hyperproliferation phenotype, the former being more severe and disorganized than the latter. These results further support our model that both miR-142 and miR-150 regulate proliferation.

3) The results of the *in vivo* transplantation experiments using human breast cancer stem cells (BCSCs) isolated from a human breast cancer xenograft demonstrate that the inhibition of miR-142 significantly suppressed the growth of the human breast cancer cells *in vivo*.

*Major points*:

*1) Reviewer 2's suggestions for testing your model more rigorously seems reasonable, namely 1) to knockdown miR-142 in breast cancer cells (which you've already done) or mammary epithelial cells and perform qPCR for APC and miR-150. The prediction would be APC upregulation and miR-150 down-regulation. And 2) to inhibit WNT signaling in breast cancer cells or mammary epithelial cells and test whether this leads to reduced miR-150 expression*.

In the revised manuscript, we show that the knockdown of miR-142 increases the protein level of APC and decreases the expression of miR-150 in the human breast cancer cells derived from the human breast cancer xenograft (Figure 5 and Figure 7). As for the second point, our results show that the knockdown of β-catenin suppressed the upregulation of miR-150 in the miR-142 expressing breast cancer cells (Figure 5), further suggesting that the enhancement of miR-150 expression by miR-142 is at least partially mediated by the canonical Wnt signaling.

We note that our new results show that miR-142 upregulates the expression of miR-150 when miR-142 is highly expressed and/or the Wnt signaling pathway is strongly stimulated in mammary epithelial cells. Considering that miR-142 is highly expressed in human BCSCs, but weakly expressed or undetectable in the stem/progenitor population of the mammary epithelial cells, our result suggest that the upregulation of miR-142 and its enhancement of the miR-150 expression seem to be especially relevant in the breast tumor progression *in vivo*. These issues are discussed in the Discussion section.

*2) Both reviewers 1 and 2 were troubled by the apparent discrepancy between the in vivo and in vitro data. Based on the model, miR-142 over-expression or inhibition should phenocopy miR-150 over-expression or inhibition. However, this doesn't seem to be the case as during mammary gland regeneration in vivo, miR-142 and miR-150 have opposite effects: one inhibits regeneration while the other promotes it even though in both cases, beta-catenin is stabilized and activated. It would be essential to reconcile this in vivo observation with the proposed mechanism derived from your in vitro study that both sit in the same pathway with miR-142 activating Wnt and miR-150 suppressing APC*.

In response to your previous correspondence, we have immunostained the mammary tissues regenerated by the miR-142- and miR-150-expressing mammary cells using an anti-PCNA antibody as a maker for proliferation. Our results show that both the miR-142-expressing ‘dysplastic’ mammary tissue and the miR-150-expressing ‘hyperproliferative’ mammary tissues are highly stained with an anti-PCNA antibody (Figure 6). In contrast, the cells in the control mammary glands are hardly stained with this antibody. These results show that the changes seen with the enforced expression of miR-142 are a type of hyperproliferation that is accompanied with abnormal morphology (dysplastic changes). These results suggest that the phenotypes of both the miR-142-expressing and miR-150-expressing mammary tissues are hyperproliferative and support that miR-142 and miR-150 are in a pathway. Because the word ‘dysplastic’ is a medical pathology term that the broad readership might not be familiar with, we paid careful attention and rewrote the results and discussion to describe our findings on the miR-142-expressing mammary tissues in the manuscript.

*3) In regards to point 2 above, reviewers 1 and 3 point out that there is no demonstration that the breast cells overexpressing miR-142 eventually develop tumours or are more prone to getting tumours compared to wildtype cells. In vivo evidence is mandated to demonstrate that miR-142-induced dysplasia and miR-150-induced hyperplasia is relevant to actual breast cancer and not merely a correlation of expression level seems reasonable and critical to the paper. Reviewer 3 also touches on this issue, pointing out that the paper is about breast cancer stem cells, and since the effects on tumorigenicity are implied throughout the paper, it is important to demonstrate conversely that the miR-142 inhibited breast cancer cells display reduced tumorigenic capacity. Reviewers 2 and 4 echo the lack of rigorous testing of your in vitro data in vivo (for instance the efficacy and miRNA dosage of your shRNA-based strategy to knock down miR-142 and transfected miRNA precursors to overexpress miR-142), and point out the need to address whether WNT signaling is altered in the presence of miR-142 in vivo*.

To confirm that miR-142 is relevant to the actual breast cancer and BCSCs *in vivo*, we infected the human BCSCs isolated from the human breast cancer xenograft using a flow cytometer with the anti-miR-142-expressing or control lentivirus. Then we transplanted them into NSG mice. The growth of the tumors formed by the human BCSCs transfected with the anti-miR-142-expressing lentivirus was significantly slower than those of the control tumors formed by the control lentivirus transfected BCSCs (Figure 7). We further confirmed that the protein level of APC was elevated in the human breast cancer xenograft formed by the human BCSCs transfected with the anti-miR-142-expressing lentivirus (Figure 7), which is consistent with our observations using the breast cancer cell lines.

The result of the quantitative real-time PCR showed that the expression levels of miR-142 in MCF7 and MDA-MB-231 cells transfected with miR-142-expressing lentivirus were elevated to the level which was comparable to that of human BCSCs isolated from the 11 surgically resected primary human breast cancer specimens (Ct values distribute between 23-30 when the Ct value for internal control gene small nucleolar RNA C/D box 96A was 20).

Because miRZip sequence that binds to the endogenous microRNA and prevents it from functioning do not generally cause a degradation of the endogenous microRNA, we did not observed the change of the miR-142 level by qPCR in the miR-142-3p miRZIP (anti-miR-142-3p)-expressing cells we analyzed, except for the CommaD-beta cells and the human breast cancer xenograft cells in which the amount of miR-142 in the anti-miR-142-3p-expressing lentivirus infected cells was less than one third of the control cells. We further confirmed that the protein level of APC was elevated in the human breast cancer xenograft cells transfected with the anti-miR-142-3p-expressing lentivirus than those transfected with the control lentivirus *in vivo* (Figure 7). In addition, we analyzed the transfection efficiency of the anti-miRNA lentivirus into the human BCSCs using a flow cytometer. We found that 94% and 96% of BCSCs were transfected with the control or anti-miR-142-expressing lentivirus, respectively, two days after the transfection.

These results are consistent with our *in vitro* observations presented in the manuscripts and show that miR-142-induced dysplasia and miR-150-induced hyperplasia is relevant to actual human breast cancers. These data are presented in Figure 7 and described in the Results of the manuscript.

*Reviewer #1*:

*There are a few additional issues that seem important to address. It was not clear from the study how miR-142 and miR150 levels in cancer cells (BCSC or NTG) compare to levels in normal mammary tissue. Are these microRNAs up-regulated in tumor versus normal mammary tissue*?

We previously analyzed the miRNA expression profiles of the cancer cells in the BCSC and NTG cell populations from the 11 surgically resected primary human breast cancer specimens, and the epithelial cells in the stem/progenitor cell and other epithelial cell populations of human normal breast tissue. The expression of miR-142 in the cancer cells in the BCSC population was on average about 2^4^ times higher than those in the NTG cell population. In contrast, the expression of miR-142 was undetectable in the epithelial cells in the stem/progenitor cell and the other epithelial cell populations of human normal mammary tissues.

The expression level of miR-150 in the cancer cells in the BCSCs population of the primary human breast cancers was about 2^3^ times higher than that in the epithelial cells in the human normal mammary tissues. These results are in line with our finding that miR-142 enhances the expression of miR-150.

*The current data are not sufficient to conclude that the miR-142/APC/Wnt/miR-150 pathway is specifically required by the tumor-propagating cancer cells; it could just be a mechanism to bulk tumor growth*.

Our new data show that the knockdown of miR-142 by infecting the anti-miR-142-expressing lentivirus to the human BCSCs significantly suppressed the growth of human breast cancers *in vivo* (Figure 7). In addition, the expression of APC protein was elevated in the breast cancer xenograft generated by the anti-miR-142-3p-expressing BCSCs (Figure 7). Together with our findings that the organoid forming ability by human BCSCs was significantly suppressed and the expression of miR-150 was reduced when miR-142 was selectively knocked down in human BCSCs (Figure 4 and Figure 5), our results suggest that the miR-142/APC/Wnt/miR-150 pathway is involved in the regulation of the human BCSCs. Because the expression levels of miR-142 and miR-150 are about 2^4^ and 2^3^ times lower in the non-tumorigenic cancer cells than in the tumorigenic BCSCs of human breast cancer, we speculate that the importance of the miR-142/APC/Wnt/miR-150 pathway in tumor propagation is much higher in the BCSCs than in the non-tumorigenic cancer cells.

*Finally, what is the percentage of human breast cancers with APC suppression that are not due to somatic mutation, promoter methylation or LOH? This information would help to assess whether this miRNA-mediated APC inhibition is a relevant mechanism in human cancer*.

The most recent analyses of the TCGA sequencing project from ∼962 patients’ tumors show only a couple of mutations in β-catenin or GSK3B (which degrades it). However, about ∼2% of patients have a mutation or deletion of APC (the TCGA Research Network). Previous studies reported that there is a reduction or loss of expression of the APC protein in 40.7% of primary human breast cancers (Ho et al., 1999), and it appears to be more often due to promoter methylation (36-54%) and loss of heterozygosity (LOH) (23%) than to somatically acquired APC mutations in human breast cancers (43, 25, 3, 3). However, the APC mutations in breast cancers are much more frequently seen in advanced than early stage breast cancers (18).

Wnt signaling is a regulator of the development and maintenance of the mammary tissues and the initiation and growth of breast cancers. We and others showed that miR-142 enhances the Wnt signaling and our results show that expression of miR-142 is highly upregulated in BCSCs isolated from human breast cancers. These results suggest that the stimulation of the Wnt signaling pathway is critically important for some human breast cancers especially at the initial step of the breast cancer formation.

Below, I delineate the experimental evidence that would need to be bolstered to substantiate the conclusions drawn:

Figure 1*: What is the efficiency of the miR150-dependent recruitment of specific mRNAs to the RISC complex? Does this correlate with the types of miRNA seed matches as miR-142 does? The authors should minimally show control data that the miR-150 Ago2IP was efficient*.

The efficiency of the miR150-dependent recruitment of specific mRNAs to the RISC complex correlated with the types of miRNA seed matches as miR-142 does. The data are presented in Figure 1—figure supplement 1.

*To substantiate that miR-150 does not target APC as suggested by AgoIP, the authors should use luciferase assays to confirm*.

The results of the luciferase assays show that miR-150 suppressed the activity of the luciferase expression plasmid in which the potential target site within the APC mRNA was fused with a luciferase gene by 32%. However, in contrast to miR-142, *APC* mRNA was not efficiently enriched by miR150 in the Ago IP experiment ([Supplementary-material SD1-data SD2-data SD3-data SD4-data]), and the activity of TOP Flash plasmid that contains the TCF binding site within its promoter was much weaker in the miR-150 transfected cells than in the miR-142 transfected cells. Because each miRNA has multiple target genes, we speculate that the presence of many other target genes with higher affinity to the miR-150 containing RISC will perturb the ability of miR-150 to regulate APC and miR-150 has much weaker effect, if any, on the Wnt signaling pathway *in vivo*. We touched this point in the Discussion.

Figure 2*: What is the relative miR-142 overexpression level by the pre-miR-142 treatment in MCF7 and MDA-MB-231 cells? How much knock down is achieved with miR-142-3p miRZIP? These are important measurements to show how much endogenous miR-142-3p was perturbed by the various tools the authors used*.

We responded these issues at the #3 of the Major points raised by the editors and the reviewers.

References:

1) Ho KY, Kalle WH, Lo TH, Lam WY, Tang CM 1999 Reduced expression of APC and DCC gene protein in breast cancer. *Histopathology* 35: 249-256.

2) Sarrio D, Moreno-Bueno G, Hardisson D, Sanchez-Estevez C, Guo M, Herman JG, et al. 2003 Epigenetic and genetic alterations of APC and CDH1 genes in lobular breast cancer: relationships with abnormal E-cadherin and catenin expression and microsatellite instability. *Int J Cancer* 106: 208-215.

3) Jin Z, Tamura G, Tsuchiya T, Sakata K, Kashiwaba M, Osakabe M, et al. 2001 Adenomatous polyposis coli (APC) gene promoter hypermethylation in primary breast cancers. *Br J Cancer* 85: 69-73.

4) Banerji S, Cibulskis K, Rangel-Escareno C, Brown KK, Carter SL, Frederick AM, et al. 2012 Sequence analysis of mutations and translocations across breast cancer subtypes. *Nature* 486: 405-409.

5) Furuuchi K, Tada M, Yamada H, Kataoka A, Furuuchi N, Hamada J, et al. 2000 Somatic mutations of the APC gene in primary breast cancers. *Am J Pathol* 156: 1997-2005.

*Reviewer #2*:

*1) If the above pathway is the model presented, a simple experiment would be to knockdown miR-142 in breast cancer cells (which the authors have already done) or mammary epithelial cells and perform qPCR for APC and miR-150. Prediction would be APC upregulation and miR-150 down-regulation. Have the authors done this*?

*2) A second prediction would be that inhibiting WNT signaling in breast cancer cells or mammary epithelial cells would lead to reduced miR-150 expression. This should be tested*.

We appreciate the reviewer for these important comments. We responded these issues at the #1 of the Major points raised by the editors and the reviewers.

*3) One thing that I don't understand is that based on the above model, miR-142 over-expression or inhibition should phenocopy miR-150 over-expression or inhibition. However, this doesn't seem to be the case as*
Figure 6
*reveals opposite effects for miR-142 and miR-150. How do the authors reconcile this*?

We appreciate the reviewer for these important comments. We responded these issues at the #2 of the Major points raised by the editors and the reviewers.

*4) Have the authors tried to rescue the miR-142 organoid phenotype by introducing the APC coding sequence and determining if the effect is reversed*?

We used the organoid culture system that is able to recapitulate the function of stem/progenitor cells and their differentiation of the primary cells derived from normal or cancer tissues. However, it is practically difficult to perform the rescue experiments in the organoid culture system because the lentivirus transfection is required for the gene transduction into the primary stem/progenitor cells and the APC gene (> 8.5k base pairs) is too large to be cloned into the lentivirus.

*5) Since the paper is about breast cancer stem cells and effects on tumorigenicity are implied throughout the paper, it is important for the authors to demonstrate that the miR-142 inhibited breast cancer cells display reduced tumorigenic capacity*.

We appreciate the reviewer for these important comments. We responded these issues at the #3 of the Major points raised by the editors and the reviewers.

*6) The authors should include control experiments on proliferation and apoptosis experiments upon miR-142 or miR-150 over-expression. I presume that there shouldn't be an effect since the effect is on renewal*?

Our new data show that cell proliferation of the mammary tissues regenerated by the miR-142-expressing and miR-150-expressing mammary cells was enhanced compared to those regenerated by the control mammary cells (Figure 6). The cell proliferation was suppressed and apoptosis was enhanced in the anti-miR-142 transfected human breast cancer xenograft cells (Figure 4). The regulation of cell proliferation and the suppression of apoptosis are a part of the self-renewal abilities that characterizes stem cells. We have previously shown that BMI1, a Polycomb group gene that has critical roles in the self-renewal of various stem cells, regulates proliferation and apoptosis by regulating the expression of the p16^Ink4a^ and p19^Arf^ gene locus (Park, et al., 2003, Nature, *423*, 302-305). In this manuscript, we presented that miR-142 targeted APC, activated the Wnt signaling pathway, and induced the miR-150 expression. Because Wnt signaling is involved in the regulation of proliferation and apoptosis, and miR-150 is reported to regulate proliferation of breast cancer cells, we think that the functions of these microRNAs include the regulation of cell proliferation and apoptosis.

*Reviewer #3*:

*1) In*
Figure 2*, the authors use sh-RNAs against miR-142 in HEK293T and MDA-MB-231 cells, and they observe downregulation of APC protein levels. As previously described and cited by the authors herein, the MDA-MB-231 and HEK293T cells express low levels of miR-142. How anti-miR-142 can affect the APC levels in these cell lines if they don't express this miRNA? The authors should use another cell line, expressing miR-142 at a significant level, in order to prove that it is indeed the specific activity of sh-miR-142 that is responsible for the downregulation of APC. Showing the levels of miR-142 before and after the use of shRNA is also essential*.

We appreciate the reviewer for these important comments. We responded these issues at the #3 of the Major points raised by the editors and the reviewers.

*2) With reference to*
Figure 2*, there is a lack of mechanistic insight into how miR-142 inhibits organoid formation in BCSCs. While the authors have demonstrated that miR-142 stimulates canonical WNT signaling via the inhibition of APC in HEK293T, MCF7 and MDA-MB231 cell lines, the role of miR-142 in the regulation of WNT signaling in the CSC population of MMTV-Wnt1 tumours was not formally tested. What is the level of the APC protein and TOP/FOPflash readout in sh-miR-142 infected cells*?

The transduction of the genes into the primary cells requires the lentivirus and the transduction is much less efficient. Although we could not perform TOP/FOPflash experiments in the MMTV-Wnt1 tumors using the lentivirus with the TOP/FOPflash constructs, we evaluated the effect of anti-miR-142-3p using the human breast cancer xenograft cells. The result of the real-time PCR analyses show that the amount of miR-142 in the anti-miR-142-3p transfected cells was less than one third of the control cells. We further confirmed that the protein level of APC was elevated in the human breast cancer xenograft cells transfected with the anti-miR-142-3p lentivirus (Figure 7). These results suggest that anti-miR-142-3p is efficiently working and miR-142 targets APC in the primary breast cancer cells.

*3) In*
Figure 4
*the authors present the effect of anti-miR-142 to spheroid formation to BCSC from mouse and human mammary tumours. The authors miss a control here, which is same culture setting with the addition of a Wnt inhibitor (ShRNA against b-catenin)*.

The requested experiment is impossible to perform because the primary normal and many cancerous mammary cells fail to form organoids in the absence of Wnt agonists.

*4) The authors state that their results suggest that miR-142 and miR-150 regulate Wnt signaling in BCSCs. However, the only relevant data in the present manuscript is the spheroid-forming assay using shRNA against miR-142, without estimating the Wnt pathway activity with or without the addition of shRNA and without testing the effect of overexpression or inactivation of miR-150 in BCSC. They should perform spheroid forming assays with overexpression/deletion of both miRNAs, compare and combine these assays with Wnt pathway inhibition, and finally perform in vivo tumour forming assays*.

In this manuscript, we focused the role of miR-142, one of the miRNAs upregulated in human BCSCs, and found that miR-142 targets APC, up-regulates the Wnt signaling, and induced the expression of miR-150 that has an ability to induce hyperproliferation of mammary tissues. Thus, we think that the roles of miR-150 in the organoid forming activities are interesting, but not critical for understanding our results.

We evaluated the effect of anti-miR-142-3p using the human breast cancer xenograft cells and confirmed that the protein level of APC was elevated in the human breast cancer xenograft cells transfected with the anti-miR-142-3p-expressing lentivirus (Figure 7). Finally, we performed the *in vivo* tumor forming assays using the anti-miR-142-3p-expressing human BCSCs and the results are presented at the response to the major points raised by the editors and the reviewers #3.

*5) From the results presented in*
Figure 5*, the authors conclude in the corresponding text that miR-142 upregulates miR-150 via the canonical Wnt signaling. To prove this point, the authors should use miR-142 expressing cells and show that the levels of miR-150 decrease upon addition of shRNA against β-catenin*.

We confirmed that the protein level of APC was elevated and the expression of miR-150 was decreased in the human breast cancer xenograft cells transfected with the anti-miR-142-3p lentivirus (Figure 5 and Figure 7). And the transfection of the siRNA against β–catenin resulted in the reduction of miR-150 expression in the miR-142-3p-expressing human breast cancer cells (Figure 5).

*6) More importantly, the*
Figure 6
*shows that when mammary epithelial cells are transfected with miR-150 and grafted in empty mammary fat pads, they recapitulate the phenotype of activated Wnt signaling (largely hyperplastic mammary gland). However, when transfected with miR-142, they observe the opposite phenotype (complete destruction of the mammary ducts), which clearly does not phenocopy the previously reported phenotypes of Wnt gain of function in the mammary gland. If miR-142 activates both Wnt signaling and miR-150 expression, then miR-142 transfection should give a comparable phenotype. These results therefore contradict the model derived from the in vitro data, and indicate that maybe this is not the case in mammary epithelial SC. Could miR-142 be regulating another key player that may lead to the observed dysplasia? Can the phenotype in miR-142 transduced and grafted cells be rescued with the knockdown of miR-150? The authors should also test whether WNT signaling is changed in this context by crossing with WNT-signaling reporter mice (for example TOPGAL or bat-gal)*.

In the original manuscript, we were not clear that both miR-142 and miR-150 cause a hyperproliferation defect. In the revised manuscript, we provide additional evidence that clearly shows that increased expression of both miR-142 and miR-150 induce proliferation. We are sorry that we did not make this point clearer in the original manuscript.

*7) The experiments presented in*
Figure 6
*should be repeated using the CSC population in MMTV-Wnt-1 tumours and human BCSCs in order to substantiate the claims of the authors made in the title of the manuscript*.

We appreciate the reviewer for these important comments. We responded these issues at the #3 of the Major points raised by the editors and the reviewers.

*8) The β-catenin staining presented in*
Figure 7
*is not convincing. Immunofluorescence of β-catenin is known to give very high background and false positive results, and the assay of choice to reliably test the expression of nuclear b-catenin is immunohistochemistry with proper controls*.

In the revised manuscript, we present the result as Figure 6—figure supplement 1. As the reviewer pointed out, the immunohistochemistry of β-catenin in general suffers from nonspecific staining. However, in the experiments presented in Figure 6—figure supplement 1, we detected the strong positive signal in the dysplastic mammary tissue regenerated by the miR-142-expressing mammary cells. In contrast, it is difficult to detect the β-catenin signals in the mammary tissue regenerated by the mammary cells transfected with the control lentivirus. We speculate that nuclear β-catenin was detectable in the immunohistochemistry experiments presented in Figure 6—figure supplement 1 because the Wnt signaling pathway was highly overstimulated in the miR-142-expressing mammary cells. We described these issues in the Result section.

*Reviewer #4*:

*1) The regulation by microRNAs is highly sensitive to cellular context. It was not clear why the authors performed their elegant Ago/IP experiments in HEK293T cells with transfected miR-142. In this experiment, both mRNA context and the expression of miR-142 were artificial (*Figure 1*). The result would yield little insights for how miR-142 would function in BCSCs. Although the authors subsequently validated the targeting of APC by miR-142 in MCF7, MDA-MB-231 cells, these cells also don't present BCSCs as identified by the authors (*Figure 2*). Thus, it remains a major task to confirm that APC is indeed regulated by miR-142 in BCSCs and determine the level of regulation in this context. In addition, it is now widely accepted that an miRNA can target hundreds of mRNAs. It is unclear to me why APC stands out as a target of particular importance, other than its well-documented role in the WNT signaling. It is not sufficiently explored whether the regulation of APC is a major or minor function of miR-142. As such, it remains an open possibility that miR-142 regulates other genes also involved in the WNT activation or other completely different pathways. Without these studies, it remains unclear whether miR-142 indeed functions through its control of the WNT signaling*.

We completely agree that there are likely other important miR-142 targets. However, our data clearly suggests that the β-catenin pathway is an important target for miR-142.

The second part of the comment questions the importance of APC and the Wnt signaling as the target of miR-142 in breast cancer and BCSCs. The Wnt signaling is a critical regulator of the development and maintenance of the mammary tissues and the initiation and growth of breast cancers. In addition to the fact that APC is among the top list of the comprehensive Ago IP/microarray experiments, our results of the luciferase assays, Western blots including the analyses using the anti-miR-142-3p, and TOP Flash experiments, indicate that miR-142 strongly upregulate the activity of the canonical Wnt signaling pathways. However, miRNAs has hundreds of the predicted targets in general and there remains a possibility that other pathways have roles in the formation of the phenotypes

*2) The authors used an shRNA-based strategy to knock down miR-142 and transfected miR precursors to overexpress miR-142. The function of miRNAs is highly sensitive to its dosage (Mukherji S et al., 2011). Thus, it is critical to document the efficacy and miRNA dosage with such experimental manipulations. One could argue that the overexpression experiment introduces a very high level of miR-142 and generates artifacts due to the high levels. The authors, at minimum, should compare the levels of overexpressed miR-142 to the elevated miR-142 observed in BCSCs. On the other hand, the authors should also document the knock down efficiency by the means of Northern blotting, qPCR or reporter assays esp. for a knock-down method that is not very well documented*.

We appreciate the reviewer for these important comments. We responded these issues at the #3 of the Major points raised by the editors and the reviewers.

*3) In*
Figure 3*, the authors performed an interesting experiment to overexpress APC with WT and the mutated miR-142 target site in the presence of miR-142. However, the 3'UTR used was not the full length but a fragment. It is demonstrated that the position of miRNA target sites could affect the strength of regulation (Grimson A et al., 2007). Thus, by using a shortened 3'UTR, the authors could potentially alter the 3'UTR context of the miR-142 target site that, in turn, leads to artifacts. They should use the full-length 3'UTRs in all assays*.

We agree the reviewer that position of the target site affected the strength of the regulation. However, the APC mRNA with the full length 3’UTR (>10k base pairs) was too long to be evaluated in the transfection assays. We hope that our new data using the anti-miR-142-expressing cells further support the roles of miR-142 on the regulation of the APC protein expression.

*4) In*
Figure 4*, the authors performed a series of experiments to document to the role of miR-142 in mouse and human BCSCs. However, the characterization is superficial and limited to simple organoid formation assay. Is it caused by reduced proliferation and/or enhanced cell death and/or cell senescence? Is the repressed WNT signaling by miR-142 knock down responsible for such defects or other pathways were also involved? More careful analysis is warranted for this study*.

We analyzed the proliferation and apoptosis of the organoid formation by human BCSCs. We found that the cell proliferation was suppressed and apoptosis was enhanced in the anti-miR-142 transfected human breast cancer xenograft cells (Figure 4).

Our new data show that the knockdown of miR-142 in the human breast cancer xenograft cells resulted in the increase of the protein level of APC and decrease of the miR-150 expression (Figure 5 and Figure 7). In addition, the transfection of the siRNA against β-catenin resulted in the reduction of miR-150 expression in the miR-142-expressing human breast cancer cells (Figure 5). And we confirmed that the growth of the tumors formed by the human BCSCs transfected with the anti-miR-142-expressing lentivirus was significantly slower than those of the control tumors formed by the control lentivirus transfected BCSCs (Major points raised by the editors and the reviewers #3) These data suggest that the regulation of APC and the Wnt signaling is at least one of the important pathways targeted by miR-142 in human breast cancer cells and BCSCs.

*5) In*
Figure 6*, the authors carried out mammary gland regeneration assay to probe the functions of miR-142/150. However, it is not explained whether miR-142/150 are up-regulated in normal mammary gland or during mammary gland regeneration or during tumor initiation of breast cancer. Without such information, the defective mammary gland regeneration caused by overexpression of miR-142/150 provides little insights for the roles of these miRNAs in BCSCs*.

We analyzed the miRNA expression profiles of the cancer cells within the BCSCs and NTG cell populations from the 11 surgically resected primary human breast cancer specimens, and the epithelial cell within the stem/progenitor cell and other epithelial cell populations of human normal breast tissue. The expression of miR-142 was significantly upregulated in the cancer cells within the BCSC population, which was about 2^4^ times higher than those in NTG cancer cells. In contrast, the expression of miR-142 was undetectable in the epithelial cells within the stem/progenitor cell and the other epithelial cell population of human normal mammary tissues. The expression of miR-150 was observed in human normal mammary tissues. The expression level was about 2^3^ times upregulated in the cancer cells within the BCSC population of the primary human breast cancers. These results support our hypothesis that miR-142 which is significantly unregulated in human BCSCs enhances the expression of miR-150 in human breast cancer cells.

*[Editors’ note: further clarifications were requested prior to acceptance, as outlined below*.*]*

*1) The reviewers asked the authors to understand the discrepancy between the in vivo and in vitro data. Based on the model, miR-142 over-expression or inhibition should phenocopy miR-150 over-expression or inhibition. However, this doesn't seem to be the case as during mammary gland regeneration in vivo, miR-142 and miR-150 have opposite effects: one inhibits regeneration while the other promotes it even though in both cases, beta-catenin is stabilized and activated. It would be essential to reconcile this in vivo observation with the proposed mechanism derived from the in vitro study that both sit in the same pathway with miR-142 activating Wnt and miR-150 suppressing APC. The authors respond to this comment by saying that different micro-RNAs may regulate many different functions, which is probably true, but it does not help to understand their data and proposed mechanisms if they are not substantiated by their functional experiments in vivo*.

We appreciate the suggestions. In this manuscript, we showed that miR-142 enhanced the activity of the Wnt signaling pathway *in vitro* and the nuclear localization of active β-catein *in vivo*. And our result of the chromatin immunoprecipitation experiments showed that the promoter region of the miR-150 precursor was bound by β-catenin. To address your concerns, we will perform the following experiments:

1) First, we will treat MDA-MB-231 and/or MCF7 breast cancer cells with either LiCl or Wnt3A in order to activate the Wnt signaling pathway. Expression of miR-150 will be measured using qPCR.

2) Next, the breast cancer cells will be transduced with an shRNA targeting β-catenin and expression of miR-150 will be measured.

3) Finally, the breast cancer cells will be transduced with an siRNA or shRNA targeting miR-142 or a control siRNA/shRNA, with or without stimulating the cells with LiCl or Wnt3A in order to activate the Wnt signaling pathway, and qPCR for APC and miR-150 will be done to measure their expression.

These experiments should address the referees’ concerns.

*2) The revised version will need a better explanation of how the authors reconcile the discordant phenotypes of miR-142 over-expression/inhibition with miR-150 over-expression/inhibition given that they are proposing these miRNAs are in a pathway*.

We are sorry that we failed to convey our results regarding the effects of enforced expression of miR-142 and miR-150 on normal mammary gland biology more clearly. Our current model is that miR-142 enhance the Wnt signaling pathway by reducing the APC protein level, and induce the transcription of miR-150 as one of the Wnt target genes.

Both miR-142 and miR-150 cause aberrant proliferation of the mammary cells *in vivo*. However, dysplastic changes were seen in miR-142 expressing cells while enforced expression of miR-150 only resulted in hyperproliferative changes. As reviewer 2 pointed out, these results are consistent with previous observations using a breast cancer cell line that miR-150 has the ability to promote cell proliferation. It is not surprising that miR-150 overexpression does not fully recapitulate the effects of miR-142 expression. The Blelloch group has previously shown that microRNAs regulate multiple targets each of which partially contributes to the cellular processes controlled by the microRNA (Subramanyam D, Lamouille S, Judson RL, Liu JY, Bucay N, Derynck R, Blelloch R. Multiple targets of miR-302 and miR-372 promote reprogramming of human fibroblasts to induced pluripotent stem cells. Nat Biotechnol. 2011 May; 29(5):443-8.).

*3) It is well recognized that a single miRNA can target hundreds of targets, and miR-142 has been shown to target APC previously. The authors' central hypothesis is that miR-142's effect in breast cancer is mediated through the activation of the Wnt pathway that activates miR-150 as a result. This discrepancy should be experimentally addressed in the revised version*.

We agree that this data would substantially strengthen the manuscript. We have performed the first batch of the experiments in which miR-142 was knocked down using a sh-miR-142-expressing lentivirus in the breast cancer stem cells derived from the human breast cancer xenograft. These cells were transplanted into immunodeficient mice. The results show that the shRNA knockdown of miR-142 prolongs the time to initial engraftment and inhibits the tumor growth of human breast cancers generated by the transplantation of human breast cancer stem cells. Thus the result is promising, but we need to repeat them to be sure that the results are correct. Because induction of the genes or shRNAs into the breast cancer stem cells are possible by using the lentivirus and the growth of the human breast cancer cells in the mouse xenograft models takes up to three months, we think that it will take three to four months to replicate this experiment.